# KAA: Kolmogorov-Arnold Attention for Enhancing Attentive Graph Neural Networks

**Taoran Fang**
Zhejiang University
`fangtr@zju.edu.cn`

**Tianhong Gao**
Shanghai Jiaotong University
`gth2002@sjtu.edu.cn`

**Chunping Wang**
FinVolution Group
`wangchunping02@xinye.com`

**Yihao Shang**
Zhejiang University
`shang_yihao@zju.edu.cn`

**Wei Chow**
Zhejiang University
`3210103790@zju.edu.cn`

**Lei Chen**
FinVolution Group
`chenlei04@xinye.com`

**Yang Yang**[*]
Zhejiang University
`yangya@zju.edu.cn`

## Abstract

Graph neural networks (GNNs) with attention mechanisms, often referred to as attentive GNNs, have emerged as a prominent paradigm in advanced GNN models in recent years. However, our understanding of the critical process of scoring neighbor nodes remains limited, leading to the underperformance of many existing attentive GNNs. In this paper, we unify the scoring functions of current attentive GNNs and propose Kolmogorov-Arnold Attention (KAA), which integrates the Kolmogorov-Arnold Network (KAN) architecture into the scoring process. KAA enhances the performance of scoring functions across the board and can be applied to nearly all existing attentive GNNs. To compare the expressive power of KAA with other scoring functions, we introduce Maximum Ranking Distance (MRD) to quantitatively estimate their upper bounds in ranking errors for node importance. Our analysis reveals that, under limited parameters and constraints on width and depth, both linear transformation-based and MLP-based scoring functions exhibit finite expressive power. In contrast, our proposed KAA, even with a single-layer KAN parameterized by zero-order B-spline functions, demonstrates nearly infinite expressive power. Extensive experiments on both node-level and graph-level tasks using various backbone models show that KAA-enhanced scoring functions consistently outperform their original counterparts, achieving performance improvements of over 20% in some cases. Our code is available at `https://github.com/zjunet/KAA`.

## 1 Introduction

Graph neural networks (GNNs) have achieved great success in graph data mining (Kipf & Welling, 2017; Hamilton et al., 2017; Xu et al., 2019; Wu et al., 2019; Sun et al., 2022) and are widely applied to various downstream tasks, such as node classification (Jiang et al., 2019a; Huang et al., 2024), link prediction (Kipf & Welling, 2016; Huang et al., 2025), vertex clustering (Ramaswamy et al., 2005; Sun et al., 2024), and recommendation systems (Ying et al., 2018). To further enhance the expressive power of GNNs, attentive GNNs (Sun et al., 2023; Chen et al., 2024) incorporate an attention mechanism (Vaswani et al., 2017) into GNN models, allowing them to adaptively learn the aggregation coefficients between a central node and its neighbors. This capability grants attentive GNNs greater expressive power and superior theoretical performance in various downstream tasks.

Despite their theoretical advantages, the practical performance of existing attentive GNNs in real-world tasks often falls short of expectations. Even with larger parameter sizes and more flexible

---

[*]Corresponding author.

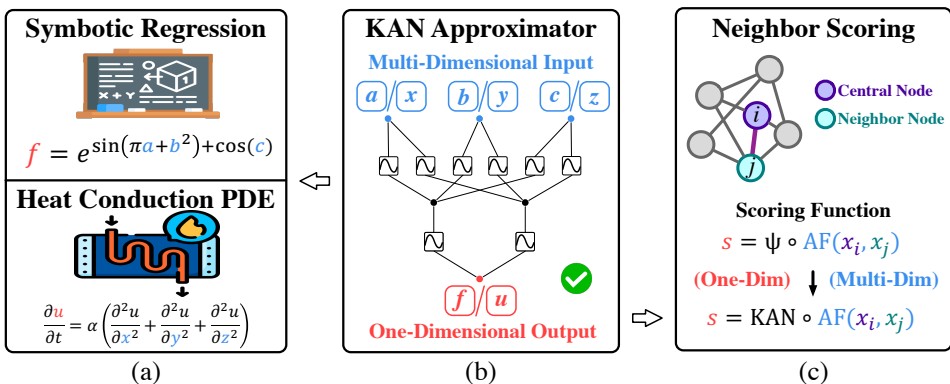

Figure 1: The alignment of our proposed KAA and other applications of KAN. (a) Symbiotic regression and PDE-solving tasks, where KAN achieves strong performance. (b) These tasks utilize KAN to handle multi-dimensional inputs and one-dimensional outputs. (c) Since the score mapping in attentive GNNs follows a similar form, we replace it with KAN.

architectural designs, attentive GNNs sometimes underperform compared to classical GNNs on certain datasets. Previous studies (Qiu et al., 2018; Brody et al., 2021) suggest that this discrepancy arises from the overly simplistic design of scoring functions in attentive GNNs, which introduces substantial inductive bias and limits their expressive power. Specifically, Brody et al. (2021) analyzed the scoring function in GAT (Veličković et al., 2018) and discovered that all central nodes tend to share the same highest-scoring neighbor, leading to subpar performance in certain scenarios. To address this, they introduced the concepts of *static* and *dynamic attention* to assess the expressive power of attentive GNN models. However, with the rapid evolution of attentive GNNs, we have observed that these analyses are primarily applicable to models with simpler structures, and their methods for evaluating the expressive power of scoring functions are too coarse, lacking the ability to provide a more granular, quantitative comparison.

To improve the effectiveness of attentive GNNs and deepen our understanding of them, two critical issues need to be addressed: how to quantify the expressive power of attentive GNN models, and how to universally enhance this expressive power. Tackling these challenges is far from straightforward. First, attentive GNN models typically involve multiple coupled linear transformations and MLPs, making it difficult to accurately quantify their theoretical expressive power. Second, the diversity in existing attentive GNN architectures makes it challenging to find a unified solution.

In our paper, we tackle these two fundamental issues from a fresh perspective. To analyze existing attentive GNN models, we first present a unified framework for their scoring functions, which consists of a learnable score mapping and a non-learnable alignment function. The scoring functions of nearly all attentive GNN models, including both GAT-based and Transformer-based approaches (Sun et al., 2023), conform to this paradigm.

To address the first issue, we introduce Maximum Ranking Distance (MRD) to quantitatively evaluate the expressive power of scoring functions. Specifically, MRD computes the upper bound of error in ranking node importance, with smaller MRD values indicating stronger expressive power. Additionally, we observe that the expressive power of the MLP module within scoring functions is often overestimated. Although the MLP possesses a universal approximation theorem (Cybenko, 1989; Hornik, 1991), in practice, limitations in the number of layers and hidden units constrain its actual expressive power. In such cases, the MRD of the scoring function with the MLP can be calculated to quantify its expressive capability.

To address the second issue, we introduce Kolmogorov-Arnold Network (KAN) into the scoring functions of existing attentive GNNs. KAN (Liu et al., 2024b;a) is an emerging network architecture that shows promise as an alternative to traditional MLPs. While existing MLP models optimize the summation coefficients between neurons during training, KAN focuses on optimizing the mapping between neurons. This novel architecture has garnered significant interest among researchers (Genet & Inzirillo, 2024; Abueidda et al., 2024; Bozorgasl & Chen, 2024; Bresson et al., 2024). Inspired by the remarkable success of KAN in symbolic regression (Ranasinghe et al., 2024;

Seguel et al., 2024; Shi et al., 2024) and PDE solving (Wang et al., 2024; Rigas et al., 2024; Wu et al., 2024a), we recognize its substantial potential for modeling mappings with multi-dimensional inputs and one-dimensional outputs, as illustrated in Figure 1. Similarly, the scoring function of attentive GNN models operates as a function with multi-dimensional inputs (representations) and a one-dimensional output (score). Consequently, we propose **K**olmogorov-**A**rnold **A**ttention (KAA), which replaces the score mapping in existing scoring functions with KAN. KAA can be adapted to nearly all scoring functions of attentive GNNs. With a comparable number of parameters, KAA achieves a lower Maximum Ranking Distance (MRD) compared to other linear transformation-based and MLP-based scoring functions, indicating that KAA possesses stronger expressive power. When the KAN utilized in KAA adheres to a specific structure, it can achieve almost any score distribution, even with a very limited number of parameters, demonstrating that KAN is particularly well-suited for scoring functions. Furthermore, we validate the effectiveness of KAA across various attentive GNN backbone models. The experimental results align with our theoretical findings, showing that KAA-enhanced scoring functions consistently outperform the original scoring functions across all tasks and backbone architectures.

Overall, the contributions of our work can be summarized as follows:

- We introduce a unified form for the scoring functions of attentive GNN models and propose Maximum Ranking Distance (MRD) to quantitatively measure their expressive power.

- We propose Kolmogorov-Arnold Attention (KAA), applicable to nearly all attentive GNN models. KAA integrates a KAN learner into the scoring function, significantly enhancing its expressive power compared to linear transformation-based and MLP-based attention in practical scenarios.

- We conduct extensive experiments to validate the effectiveness of KAA across various backbone models and downstream tasks. The results demonstrate that KAA-enhanced models consistently outperform their original counterparts across all node-level and graph-level tasks, with performance improvements exceeding 20% in some cases.

## 2 RELATED WORK

### 2.1 ATTENTIVE GRAPH NEURAL NETWORKS

GNNs have garnered significant interest and been widely adopted over the past few years. However, most models (e.g., GCN (Kipf & Welling, 2017), GraphSAGE (Hamilton et al., 2017), and GIN (Xu et al., 2019)) assign equal importance to each neighbor of the central node, which limits their ability to capture the unique local structures of different nodes. In response to this limitation, GAT (Veličković et al., 2018) was the first to introduce a simple layer with an attention mechanism to compute a weighted average of neighbors' representations. The strong performance of GAT across various downstream tasks has sparked widespread interest among researchers in attentive GNN models. Attentive GNNs can be broadly categorized into two main types: GAT-based models (Veličković et al., 2018; Kim & Oh, 2021) and Graph Transformers (Nguyen et al., 2022; Zhang et al., 2020). GAT-based models assign varying weights to nodes during the feature aggregation process based on their respective influences. Since the introduction of GAT, numerous variants have emerged, such as C-GAT (Wang et al., 2019), GATv2 (Brody et al., 2021), and SuperGAT (Kim & Oh, 2021). Additionally, many models (Jiang et al., 2019a; Cui et al., 2020; Yang et al., 2021; Lin et al., 2022; Zhang & Gao, 2021) have modified how GAT combines node pair representations and are also categorized as GAT-based models. Graph Transformers (Rong et al., 2020) are a class of models based on Transformers (Vaswani et al., 2017), which can directly learn higher-order graph representations. In recent years, Graph Transformers have rapidly advanced in the field of graph deep learning. These models (Dwivedi & Bresson, 2020; Kreuzer et al., 2021; Ying et al., 2021; Xia et al., 2021; Shi et al., 2020; Fang et al., 2024) typically employ a query-key-value structure similar to that of Transformers and demonstrate superior performance on graph-level tasks.

### 2.2 KOLMOGOROV–ARNOLD NETWORKS

Kolmogorov–Arnold Network (KAN) (Liu et al., 2024b;a) is a novel neural network architecture inspired by the Kolmogorov-Arnold representation theorem (Kolmogorov, 1957; Braun & Griebel,

2009), designed as an alternative to MLP. Unlike MLP, KAN does not utilize linear weights. Instead, each weight parameter between neurons is replaced by a univariate function parameterized as a spline. Compared to MLPs, KAN demonstrates enhanced generalization ability and interpretability (Liu et al., 2024b). However, achieving good performance with KAN in practical applications can be challenging (Altarabichi, 2024a; Le et al., 2024; Altarabichi, 2024b). Drawing inspiration from KAN's remarkable success in symbolic regression (Ranasinghe et al., 2024; Seguel et al., 2024; Shi et al., 2024) and PDE solving (Wang et al., 2024; Rigas et al., 2024; Wu et al., 2024a), we recognize KAN's significant potential for fitting mappings with multi-dimensional inputs and one-dimensional outputs. Our use of KAN in this work also reflects its empirical effectiveness in these contexts.

## 3 PRELIMINARY

**Graph Data and Graph Neural Networks.** Let $\mathcal{G} = (\mathcal{V}, \mathcal{E}) \in \mathbb{G}$ represents a graph, where $\mathcal{V} = \{v_1, v_2, \ldots, v_N\}, \mathcal{E} \subseteq \mathcal{V} \times \mathcal{V}$ denote the node set and edge set respectively. The node features can be denoted as a matrix $\mathbf{X} = \{x_1, x_2, \ldots, x_N\} \in \mathbb{R}^{N \times d}$, where $x_i \in \mathbb{R}^d$ is the feature of the node $v_i$, and $d$ is the dimensionality of original node features. $\mathbf{A} \in \{0, 1\}^{N \times N}$ denotes the adjacency matrix, where $\mathbf{A}_{ij} = 1$ if $(v_i, v_j) \in \mathcal{E}$. Given a GNN model $f$, the node representation $h_i$ can be obtained layer by layer using the following expression:

$$h_i^{(l+1)} = \text{UPDATE}^{(l)}(h_i^{(l)}, \text{AGG}_{j \in \mathcal{N}(i)}^{(l)}(h_i^{(l)}, h_j^{(l)}))$$ (1)

where $h_i^{(l)}$ denotes the representation of $v_i$ at the $l$-th layer, with $h_i^{(0)} = x_i$. AGG$(\cdot)$ denotes an aggregation function, such as sum (*e.g.*, GCN) or mean (*e.g.*, GraphSAGE) aggregation. UPDATE$(\cdot)$ denotes a feature transformation function, such as a linear transformation or an MLP.

**Unified Scoring Functions in Attentive Graph Neural Networks.** Attentive GNN models introduce an attention mechanism in the aggregation function AGG$(\cdot)$ to adaptively assign different weight coefficient $\alpha_j$ to each neighboring node $v_j$, which can be expressed as:

$$\text{AGG}_{j \in \mathcal{N}(i)}(h_i, h_j) = \sum_{j \in \mathcal{N}(i)} \alpha_j h_j$$ (2)

where $\alpha_j$ is obtained by normalizing the result of the scoring function. Here, the scoring function s$(\cdot)$ calculates the importance score of a neighboring node to the central node based on their respective representations. The scoring functions of existing attentive GNN models can be categorized into two types (Sun et al., 2023): *GAT-based* and *Transformer-based*. Their representative forms of scoring functions are as follows:

$$s(h_i, h_j) = \text{LeakyReLU}(a^\top \cdot [\mathbf{W}h_i \| \mathbf{W}h_j]) \qquad \text{(GAT-based)}$$ (3)

$$s(h_i, h_j) = (\mathbf{W}_q h_i)^\top \cdot \mathbf{W}_k h_j \qquad \text{(Transformer-based)}$$ (4)

$$\alpha_j = \frac{\exp(s(h_i, h_j))}{\sum_{j' \in \mathcal{N}(i)} \exp(s(h_i, h_{j'}))} \qquad \text{(Normalization)}$$ (5)

For GAT-based scoring functions, many variants substitute the concatenation operation of the two representations with alternative operations, such as vector addition or subtraction. In contrast, Transformer-based scoring functions often introduce additional scaling factors, such as $\sqrt{\dim_{\mathrm{Q}}\dim_{\mathrm{K}}}$. We find that both types of scoring functions can be unified into the following general form:

$$s(h_i, h_j) = \Psi \circ \text{AF}(h_i, h_j)$$ (6)

where AF$(\cdot) : \mathbb{R}^d \times \mathbb{R}^d \to \mathbb{R}^{d'}$ is an alignment function without learnable parameters, such as concatenation or dot product. Its purpose is to combine the representations of the central node and the neighboring node. Additionally, $\Psi : \mathbb{R}^{d'} \to \mathbb{R}$ is the score mapping with learnable parameters, typically comprising several linear transformations and potentially some activation functions. Specifically, for the scoring function in Formula 3, the alignment function AF$(\cdot)$ is concatenation, while the score mapping $\Psi$ consists of two consecutive linear transformations followed by an activation function. For the scoring function in Formula 4, the alignment function AF$(\cdot)$ can be expressed as AF$(h_i, h_j) = h_j$, and its score mapping is given by $\Psi = (\mathbf{W}_q h_i)^\top \mathbf{W}_k$.

**Kolmogorov–Arnold Networks.** The Kolmogorov-Arnold Representation theorem (Braun & Griebel, 2009) states that, for a smooth function $f : [0, 1]^n \to \mathbb{R}$:

$$f(x_1, ..., x_n) = \sum_{q=1}^{2n+1} \Phi_q(\sum_{p=1}^{n} \phi_{q,p}(x_p)) \tag{7}$$

where $\phi_{q,p} : [0, 1] \to \mathbb{R}$, and $\Phi_q : \mathbb{R} \to \mathbb{R}$. This formulation illustrates that multivariate functions can essentially be decomposed into a well-defined composition of univariate functions, where the combination involves only simple addition. Liu et al. (2024b) extended the above expression into a single-layer KAN, enabling it to be stacked layer by layer like a regular neural network. For a single-layer KAN $\Phi$ with an input dimension of $n_{\text{in}}$ and an output dimension of $n_{\text{out}}$:

$$\forall j \in [1, n_{\text{out}}] : x_j^{\text{out}} = \sum_{i=1}^{n_{\text{in}}} \phi_{i,j}(x_i^{\text{in}}) \tag{8}$$

where $x_i$ is the value corresponding to the $i$-th dimension of $x$, and $\phi_{i,j}$ is a learnable nonlinear function, usually parameterized by B-spline functions (Liu et al., 2024b;a) or radial basis functions (Li, 2024). At this point, KAN is considered an alternative to MLPs and is applied to various tasks.

## 4 METHODOLOGY

### 4.1 KOLMOGOROV-ARNOLD ATTENTION

Inspired by the remarkable success of KAN in symbolic regression (Ranasinghe et al., 2024; Seguel et al., 2024; Shi et al., 2024), which typically involves multi-dimensional inputs and one-dimensional outputs, as well as in PDE solving (Wang et al., 2024; Rigas et al., 2024; Wu et al., 2024a), where common PDEs such as the heat equation or wave equation also feature multi-dimensional inputs and one-dimensional outputs, we find that KAN demonstrates significant potential for fitting mappings of the form $f : \mathbb{R}^n \to \mathbb{R}$ with multi-dimensional inputs and one-dimensional outputs.

According to Formula 6, the score mapping $\Phi$ also conforms to this form. Therefore, we propose Kolmogorov-Arnold Attention (KAA), which replaces $\Psi$ in the original scoring function with KAN, expressed as:

$$\text{s}(h_i, h_j) = \text{KAN} \circ \text{AF}(h_i, h_j) \tag{9}$$

After obtaining the scores for node pairs, we compute the specific weight coefficients according to Formula 5 and perform the GNN aggregation as outlined in Formula 2. This design is highly flexible, allowing KAA to be applied to nearly all existing attentive GNN models.

**Building Multi-Head Attention.** KAA can also be extended to multi-head attention (Vaswani et al., 2017; Veličković et al., 2018), further enhancing its performance. When the number of heads is $K$, we apply $K$ independent KANs of the same scale $\{\text{KAN}^1, ..., \text{KAN}^K\}$ to obtain $K$ different weight coefficients according to Formula 9 and 5. The resulting $K$ representations are then concatenated to form the final representations.

### 4.2 COMPARISON OF EXPRESSIVE POWER

In this section, we analyze the expressive power of different scoring functions. The primary purpose of designing the scoring function is to enable the model to adaptively assign varying aggregation coefficients to different nodes. This means that the scoring function should be capable of ranking the importance of neighboring nodes, allowing the central node to receive more valuable information. According to the unified form of scoring functions presented in Formula 6, $\text{AF}(h_i, h_j)$ is derived from a predefined alignment function $\text{AF}(\cdot)$ and the inherent inputs ($h_i$ and $h_j$), which cannot be modified during training. Consequently, the expressive power of the scoring function hinges on whether the learnable score mapping $\Psi$ can effectively map the various $\text{AF}(h_i, h_j)$ values to any desired importance ranking. We conduct a quantitative analysis and comparison of the expressive power of different forms of scoring functions. First, we examine the limitations of existing standards used to evaluate the expressive power of score functions. Next, we introduce a more comprehensive evaluation metric: *Maximum Ranking Distance*. Finally, we quantitatively compare the maximum

ranking distance of various scoring functions in practical scenarios. Through rigorous analysis, we demonstrate that the application of KAA can significantly enhance the practical expressive power of the scoring function.

### 4.2.1 LIMITATION OF EXISTING MEASUREMENT

Many existing works (Qiu et al., 2018; Brody et al., 2021) have explored the expressive power of scoring functions in attentive GNNs. Brody et al. (2021) initiated this line of inquiry by analyzing the scoring function of GAT and proposed two standards for evaluating the expressive power of scoring functions: *static* and *dynamic attention*. Specifically, if the scoring function assigns the highest score to the same neighboring node (key) for all central nodes (queries), it is said to compute static attention. On the other hand, if the scoring function can assign different neighboring nodes (keys) the highest score for different central nodes (queries), it computes dynamic attention. More formal and detailed definitions of these two attention types are provided in Appendix A.2. Clearly, scoring functions that compute dynamic attention have stronger expressive power. However, applying this standard to evaluate the scoring functions of various existing attentive GNNs presents several challenges. First, most existing scoring functions already compute dynamic attention, making it difficult to differentiate their expressive power using this criterion. Second, scoring functions that compute static attention can easily be adjusted to compute dynamic attention. For example, the GAT scoring function in Formula 3 initially computes static attention, but by modifying it to the following form, it can be converted to compute non-static attention:

$$\mathrm{s}(h_i, h_j) = -\mathrm{LeakyReLU}(\mathrm{Abs}(a^\top \cdot [\mathbf{W}h_i \| \mathbf{W}h_j])) \tag{10}$$

where we only add a negative sign and an absolute value function $\mathrm{Abs}(\cdot)$. These limitations make the existing measurements for evaluating various scoring functions overly simplistic and imprecise.

### 4.2.2 DESIGNING COMPREHENSIVE MEASUREMENT

We observe that existing measurements are too coarse because they focus solely on the neighbor node with the highest score. In reality, the scoring function assigns scores to all neighboring nodes, forming an *importance ranking* of the neighbors, which we define as follows:

**Definition 1** (Importance Ranking). *Given a scoring function* $\mathrm{s}(\cdot)$*, a central node with representation* $h_i$ *and all its neighbor nodes with representations* $\{h_j | j \in \mathcal{N}(i)\}$*, an importance ranking* $\sigma$ *is a permutation of* $|\mathcal{N}(i)|$*, where* $\sigma$ *is a bijective mapping* $\sigma : \{1, ..., |\mathcal{N}(i)|\} \rightarrow \{1, ..., |\mathcal{N}(i)|\}$ *that satisfies:*

$$\mathrm{s}(h_i, h_{\sigma(1)}) \leq \mathrm{s}(h_i, h_{\sigma(2)}) \leq \cdots \leq \mathrm{s}(h_i, h_{\sigma(|\mathcal{N}(i)|)}) \tag{11}$$

In contrast to static and dynamic attention, which only focus on identifying the most important neighbor, importance ranking captures the relative importance of all neighboring nodes, offering a more comprehensive evaluation. In practice, scoring functions may not always achieve the optimal importance ranking under ideal conditions; instead, they approximate it as closely as possible. To quantitatively assess the difference between two rankings, we define the ranking distance as follows:

**Definition 2** (Ranking Distance). *Given two rankings,* $\sigma_1$ *and* $\sigma_2$*, of N nodes, the ranking distance RD between them can be calculated using the following formula:*

$$\mathrm{RD}(\sigma_1, \sigma_2) = \sqrt{\sum_i^N (\sigma_1^{-1}(i) - \sigma_2^{-1}(i))^2} \tag{12}$$

*where* $\sigma^{-1}(i)$ *indicates the concrete rank of node i.*

In real-world scenarios, the optimal ranking can be any permutation of neighboring nodes. Therefore, the expressive power of a scoring function lies in its ability to produce any possible ranking. Building on this, we introduce the concept of maximum ranking distance, which quantitatively measures the expressive power of different scoring functions:

**Definition 3** (Maximum Ranking Distance). *Given a family of scoring functions* $\mathcal{S}$*, a central node with representation* $h_i$ *and neighbor nodes with representations* $\{h_j | j \in \mathcal{N}(i)\}$*, for any* $\mathrm{s} \in \mathcal{S}$*, we denote the obtained importance ranking as* $\sigma_\mathrm{s}$*. Meanwhile, the set of all permutations of* $|\mathcal{N}(i)|$ *is*

denoted as $\Pi = \{\pi | \pi : \{1, ..., |\mathcal{N}(i)|\} \rightarrow \{1, ..., |\mathcal{N}(i)|\}; \pi \text{ is a bijection}\}$. *The maximum ranking distance (MRD) is expressed as below:*

$$\text{MRD}(\mathcal{S}, h_i, \{h_j | j \in \mathcal{N}(i)\}) = \max_{\pi' \in \Pi} \min_{s' \in \mathcal{S}} \text{RD}(\sigma_{s'}, \pi') \tag{13}$$

### 4.2.3 Theoretical Comparison

In this section, we delve into the analysis of the MRD for various scoring functions, specifically examining three types: linear transformation-based attention, MLP-based attention, and our proposed KAA. These correspond to cases where the score mapping in Formula 6 is a linear transformation, MLP, and KAN, respectively. Without loss of generality, we assume that the selected central node is connected to all other nodes in the graph (as many graph transformers do), and that $\text{AF}(h_i, h_j) \in \mathbb{R}^d$. We denote the alignment matrix of all $\text{AF}(h_i, h_j)$ as $\mathbf{P} \in \mathbb{R}^{N \times d}$, where $N$ is the number of nodes. For analytical simplification, we assume $\mathbf{P}$ is derived from the first $d$ columns of a full-rank circulant matrix $\mathbf{C} \in \mathbb{R}^{N \times N}$, with $N = d^2$ (*i.e.*, $N \gg d$). A more detailed and specific elaboration can be found in Appendix A.4.

**Linear Transformation-Based Attention.** This type of scoring function is the most common, and the majority of existing attentive GNNs (Sun et al., 2023) can be classified under this category. Many practical implementations of scoring functions employ multiple consecutive learnable linear transformations (as seen in Formulas 3 and 4), but in theory, this does not enhance the expressive power compared to using a single fully learnable linear transformation, as multiple transformations are equivalent to a single transformation equal to their product. Furthermore, scoring functions like the one in Formula 3 often include a non-linear activation function. However, since most common non-linear activation functions are monotonic (*e.g.*, ReLU family and tanh), they do not alter the ranking of the scores. Therefore, we calculate the MRD for a single linear transformation as:

**Proposition 1** (MRD of Linear Transformation-Based Attention). *Given the alignment matrix $\mathbf{P} \in \mathbb{R}^{N \times d}$, for a scoring function in the form of $\text{s}(h_i, h_j) = \mathbf{W} \cdot \text{AF}(h_i, h_j)$, where $\mathbf{W} \in \mathbb{R}^{d \times 1}$, its MRD satisfies the following inequality:*

$$\text{MRD}(\mathcal{S}_{\text{LT}}, \mathbf{P}) \geq \sqrt{\frac{1}{12}(N^3 - N - d^3 + 3d^2 - 2d)} \tag{14}$$

*where $\mathcal{S}_{\text{LT}}$ is the set of all candidate linear transformation-based scoring functions.*

We have established a lower bound for the MRD of the scoring function when the score mapping is a linear transformation. A more detailed derivation can be found in Appendix A.5.

**MLP-Based Attention.** To address the limitations of linear transformations in scoring functions, Brody et al. (2021) utilized a multi-layer perceptron (MLP) as the score mapping. While an MLP is a universal approximator in ideal conditions (Cybenko, 1989; Hornik, 1991), its expressive power is often constrained by its limited width and depth in practical applications. Increasing the width and depth excessively can lead to challenges in model convergence and may result in significant overfitting (Oyedotun et al., 2017). To ensure consistency with the MLP size used by Brody et al. (2021), we calculate the MRD for the scoring function where the score mapping consists of a two-layer equal-width MLP:

**Proposition 2** (MRD of MLP-Based Attention). *Given the alignment matrix $\mathbf{P} \in \mathbb{R}^{N \times d}$, for a scoring function in the form of $\text{s}(h_i, h_j) = \mathbf{W}_2 \cdot (\text{ReLU}(\mathbf{W}_1 \cdot \text{AF}(h_i, h_j)))$, where $\mathbf{W}_1 \in \mathbb{R}^{d \times d}$ and $\mathbf{W}_2 \in \mathbb{R}^{d \times 1}$, its MRD satisfies the following inequality:*

$$\sqrt{\frac{1}{12}(N^3 - N - \lambda)} \geq \text{MRD}(\mathcal{S}_{\text{MLP}}, \mathbf{P}) \geq \sqrt{\frac{1}{12}((N - d)^3 - (N - d) - \lambda)} \tag{15}$$

*where $\lambda = d^3 - 3d^2 + 2d$ and $\mathcal{S}_{\text{MLP}}$ is the set of all candidate MLP-based scoring functions.*

For MLP-based scoring functions, we have established both upper and lower bounds for their MRD. A more detailed derivation can be found in Appendix A.6.

**Kolmogorov-Arnold Attention.** Finally, we calculate the MRD of our proposed KAA. The score mapping in KAA is KAN, which possesses a representation theorem similar to that of MLP. To ensure a fair comparison, we utilize a single-layer KAN with a comparable number of parameters to those in Proposition 2 as the score mapping:

**Proposition 3** (MRD of Kolmogorov-Arnold Attention). *Given the alignment matirx* $\mathbf{P} \in \mathbb{R}^{N \times d}$, *for a scoring function in the form of* $s(h_i, h_j) = \sum_{k=1}^{d} \phi_k(\text{AF}(h_i, h_j)_k)$, *where each* $\phi_k$ *is composed of* $d$ *modified zero-order B-spline functions* $\phi_k(x) = \sum_{l=1}^{d} c_{k,l} \cdot B_{k,l}^*(x)$, *its MRD satisfies the following inequality:*

$$\text{MRD}(\mathcal{S}_{\text{KAA}}, \mathbf{P}) \leq \delta, \quad \text{for } \forall \, \delta > 0 \tag{16}$$

*where* $\mathcal{S}_{\text{KAA}}$ *is the set of all candidate KAA scoring functions.*

A more detailed derivation can be found in Appendix A.7. The single-layer KAN scoring function contains $d^2$ learnable parameters, which is slightly fewer than the $(d^2 + d)$ learnable parameters in the MLP from Proposition 2. Despite having fewer parameters, KAA maintains nearly unlimited ranking capability, establishing it as the most expressive scoring function paradigm.

**Theorem 1.** *Under non-degenerate conditions, given the alignment matrix* $\mathbf{P} \in \mathbb{R}^{N \times d}$, *we have:*

$$\text{MRD}(\mathcal{S}_{\text{KAA}}, \mathbf{P}) \leq \text{MRD}(\mathcal{S}_{\text{MLP}}, \mathbf{P}) \leq \text{MRD}(\mathcal{S}_{\text{LT}}, \mathbf{P}) \tag{17}$$

From Theorem 1, we can conclude that in practical scenarios with limited parameters, our proposed KAA exhibits the strongest expressive power. In contrast, MLP-based attention also demonstrates a notable improvement in expressive capability compared to linear transformation-based attention.

## 5 EXPERIMENT

In this section, we systematically evaluate the effectiveness of KAA across various tasks, including node-level tasks such as node classification and link prediction, as well as graph-level tasks like graph classification and graph regression.

### 5.1 EXPERIMENT SETUP

**Backbone Models.** Our proposed KAA can be applied to both GAT-based scoring functions, as shown in Formula 3, and Transformer-based scoring functions, as shown in Formula 4. Consequently, we select three classical GAT-based attentive GNN models with varying alignment functions as our backbone models: GAT (Veličković et al., 2018), GLCN (Jiang et al., 2019b), and CFGAT (Cui et al., 2020). Additionally, we choose two Transformer-based models with distinct scoring functions as our backbone models: GT (Dwivedi & Bresson, 2020) and SAN (Kreuzer et al., 2021). The specific forms of the scoring functions are detailed in Table 1.

Table 1: A summary of the original scoring functions for various attentive GNN models and their KAA-enhanced counterparts. The central node representation is denoted as $h_i$, while the neighbor node representation is denoted as $h_j$.

| $s(h_i, h_j)$ | Original Version | KAA Version |
|---|---|---|
| GAT | $\text{LeakyReLU}(a^\top \cdot [\mathbf{W}h_i \| \mathbf{W}h_j])$ | $\text{KAN}([h_i \| h_j])$ |
| GLCN | $\text{ReLU}(a^\top \cdot |h_i - h_j|)$ | $\text{KAN}(|h_i - h_j|)$ |
| CFGAT | $\text{LeakyReLU}(\cos(\mathbf{W}h_i, \mathbf{W}h_j))$ | $\cos(\text{KAN}(h_i), \text{KAN}(h_j))$ |
| GT | $\frac{1}{\sqrt{d}}(\mathbf{W}_q h_i)^\top \cdot \mathbf{W}_k h_j$ | $\frac{1}{\sqrt{d}}\text{KAN}(h_i)^\top \cdot h_j$ |
| SAN | $\frac{1}{\sqrt{d(\gamma+1)}}(\mathbf{W}_q h_i)^\top \cdot \mathbf{W}_k h_j$ | $\frac{1}{\sqrt{d(\gamma+1)}}\text{KAN}(h_i)^\top \cdot h_j$ |

**Implementations.** For all models, we apply the dropout technique with dropout rates selected from $[0, 0.1, 0.3, 0.5, 0.8]$. Additionally, we utilize the Adam optimizer, choosing learning rates from $[10^{-3}, 5 \times 10^{-3}, 10^{-2}]$ and weight decay values from $[0, 5 \times 10^{-4}]$. Regarding model architecture, the number of GNN layers is selected from $[2, 3, 4, 5]$, the hidden dimension from $[8, 16, 32, 64, 128, 256]$, and the number of heads from $[1, 2, 4, 8]$. For the KAN and MLP modules, we adopt an equal-width structure across all models, with the number of layers chosen from $[2, 3, 4]$. In the KAN modules, B-spline functions serve as the base functions, with the grid size selected from $[1, 2, 4, 8]$ and the spline order chosen from $[1, 2, 3]$. We conduct five rounds of experiments with different random seeds for each setting and report the average results.

Table 2: Results of node-level tasks on various datasets. (The optimal results are in **bold**. "\" indicates out of memory limits, preventing full-batch training. "Avg Imp" calculates the average relative improvement of all KAA-enhanced models compared to their original versions.)

| Task / Model | Node Classification (Accuracy %) | | | | | | Link Prediction (ROC-AUC %) | | |
|---|---|---|---|---|---|---|---|---|---|
| | Cora | CiteSeer | PubMed | ogbn-arxiv | Photo | Computers | Cora | CiteSeer | PubMed |
| GCN | 81.99 | 71.36 | 78.03 | 68.80 | 92.90 | 89.45 | 92.02 | 91.00 | 94.62 |
| | ±0.70 | ±0.57 | ±0.21 | ±0.44 | ±0.12 | ±0.22 | ±0.73 | ±0.40 | ±0.06 |
| GraphSAGE | 81.48 | 69.70 | 77.50 | 68.68 | 93.26 | 88.84 | 91.28 | 87.65 | 91.62 |
| | ±0.41 | ±0.63 | ±0.23 | ±0.63 | ±0.34 | ±0.15 | ±0.54 | ±0.35 | ±0.23 |
| GIN | 79.85 | 69.60 | 77.89 | 66.04 | 90.36 | 85.34 | 89.27 | 85.84 | 91.31 |
| | ±0.70 | ±0.89 | ±0.39 | ±0.56 | ±0.18 | ±1.30 | ±1.34 | ±1.80 | ±1.09 |
| GAT | 82.76 | 72.34 | 77.50 | 68.62 | 92.87 | 89.76 | 91.48 | 90.28 | 94.03 |
| | ±0.74 | ±0.44 | ±0.75 | ±0.68 | ±0.32 | ±0.13 | ±0.37 | ±0.41 | ±0.10 |
| KAA-GAT | **83.80** | **73.10** | 78.60 | 69.02 | 93.46 | **90.36** | **92.67** | 91.48 | 94.87 |
| | ±0.49 | ±0.61 | ±0.50 | ±0.49 | ±0.23 | ±0.18 | ±0.35 | ±0.71 | ±0.19 |
| GLCN | 82.26 | 71.76 | 78.10 | 68.37 | 92.70 | 88.98 | 91.60 | 90.36 | 94.34 |
| | ±0.89 | ±0.53 | ±0.14 | ±0.60 | ±0.13 | ±0.10 | ±0.17 | ±0.79 | ±0.19 |
| KAA-GLCN | 83.50 | 72.78 | 78.42 | 68.80 | **93.90** | 89.44 | 92.57 | **91.93** | **95.09** |
| | ±0.70 | ±0.50 | ±0.16 | ±0.90 | ±0.17 | ±0.30 | ±0.20 | ±0.43 | ±0.17 |
| CFGAT | 81.42 | 71.52 | 78.94 | 68.34 | 93.12 | 89.34 | 91.85 | 90.49 | 94.06 |
| | ±0.70 | ±0.76 | ±0.33 | ±0.75 | ±0.17 | ±0.32 | ±0.35 | ±0.37 | ±0.06 |
| KAA-CFGAT | 83.72 | 72.66 | **79.04** | **69.59** | 93.33 | 89.71 | 92.58 | 91.45 | 94.71 |
| | ±0.61 | ±0.78 | ±0.32 | ±0.59 | ±0.18 | ±0.15 | ±0.40 | ±0.50 | ±0.08 |
| GT | 70.16 | 58.12 | 74.38 | \ | 89.65 | 80.25 | 86.37 | 81.64 | 84.63 |
| | ±1.80 | ±1.52 | ±1.23 | | ±1.54 | ±3.93 | ±0.81 | ±0.99 | ±4.32 |
| KAA-GT | 71.86 | 61.92 | 75.16 | \ | 92.82 | 89.05 | 86.97 | 82.35 | 88.38 |
| | ±1.34 | ±1.55 | ±0.80 | | ±0.22 | ±0.30 | ±0.87 | ±1.74 | ±1.32 |
| SAN | 71.72 | 63.18 | 72.44 | \ | 90.39 | 84.50 | 81.33 | 84.22 | 90.78 |
| | ±2.10 | ±1.60 | ±0.91 | | ±0.48 | ±0.94 | ±2.29 | ±2.13 | ±0.45 |
| KAA-SAN | 72.98 | 64.06 | 73.24 | \ | 90.92 | 85.42 | 86.35 | 84.23 | 91.28 |
| | ±0.48 | ±0.86 | ±1.30 | | ±0.86 | ±0.58 | ±1.15 | ±1.84 | ±0.33 |
| **Avg Imp ↑ (%)** | +1.95 | +2.39 | +0.82 | +1.01 | +1.25 | +2.73 | +2.00 | +1.00 | +1.47 |

## 5.2 Performance on Node-level Tasks

**Datasets Involved.** We conduct node classification and link prediction tasks to validate the effectiveness of KAA in node-level applications. For the node classification task, we select four citation network datasets (Sen et al., 2008; Hu et al., 2020) of varying scales: *Cora, CiteSeer, PubMed*, and *ogbn-arxiv*, along with two product network datasets (Shchur et al., 2018), *Amazon-Computers* and *Amazon-Photo*. The objective for the citation networks is to determine the research area of the papers, while the product networks involve categorizing products. For the link prediction task, we select three citation network datasets: *Cora*, *CiteSeer*, and *PubMed*, to predict whether an edge exists between pairs of nodes. More details and statistical information about these datasets can be found in Appendix B.1. Additionally, for all tasks, we employ three classical GNN models: GCN (Kipf & Welling, 2017), GraphSAGE (Hamilton et al., 2017), and GIN (Xu et al., 2019), as baselines.

**Experimental Results.** The results of KAA on node-level tasks are presented in Table 2. The experimental findings indicate that KAA-enhanced models achieve superior experimental results compared to the original models across all datasets, with an average improvement of 1.63%. This clearly demonstrates the enhancements that KAA provides to attentive GNN models in node-level tasks. For GAT-based models, none of the original attentive models outperform all non-attention models on any dataset or task. This phenomenon suggests that the additional parameters in the existing scoring functions do not effectively enhance performance. However, after applying KAA to these models, their performance improves significantly. All KAA-enhanced GAT-based models outperform non-attention models across all tasks and datasets. In contrast, Transformer-based models do not perform as well on the involved node-level tasks. In these cases, the benefits from KAA are even more pronounced. Specifically, KAA-enhanced models demonstrate an average performance improvement of 2.70% compared to the original models. Overall, KAA can universally enhance the performance of attentive GNN models on node-level tasks, with improvements that can even bridge the performance gap between different models.

Table 3: Results of graph-level tasks on various datasets. (The optimal results are in **bold**. "Avg Imp" calculates the average improvement of all KAA-enhanced models compared to original ones.)

| Task
Model | Graph Classification (Accuracy %) | | | | Graph Regression (MAE $\downarrow$) | |
|---|---|---|---|---|---|---|
| | PPI (F1) | MUTAG | ENZYMES | PROTEINS | ZINC | QM9 |
| GCN | 60.48 ±0.79 | 95.78 ±5.15 | 29.33 ±5.58 | 71.71 ±1.67 | 0.6863 ±0.0011 | 0.2715 ±0.0080 |
| GraphSAGE | 77.95 ±0.25 | 94.73 ±5.76 | 24.67 ±2.79 | 70.09 ±2.44 | 0.6173 ±0.0162 | 0.2733 ±0.0030 |
| GIN | 70.29 ±2.78 | 96.84 ±4.21 | 33.34 ±2.90 | 71.53 ±2.64 | 0.4759 ±0.0172 | 0.2524 ±0.0035 |
| GAT | 83.84 ±2.09 | 94.73 ±4.70 | 29.23 ±5.29 | 70.09 ±2.50 | 0.6711 ±0.0111 | 0.4749 ±0.0158 |
| KAA-GAT | 86.62 ±1.11 | 97.89 ±2.57 | 34.46 ±1.08 | 72.43 ±1.67 | 0.5247 ±0.0145 | 0.2124 ±0.0085 |
| GLCN | 74.86 ±1.15 | 96.84 ±4.21 | 31.35 ±4.39 | 71.17 ±1.88 | 0.6716 ±0.0031 | 0.2716 ±0.0168 |
| KAA-GLCN | 94.96 ±0.26 | **98.94** ±2.10 | 33.67 ±2.03 | 72.25 ±1.05 | 0.6351 ±0.0202 | 0.2684 ±0.0205 |
| CFGAT | 75.90 ±0.87 | 95.78 ±3.93 | 28.33 ±3.37 | 70.27 ±1.70 | 0.6779 ±0.0031 | 0.2624 ±0.0082 |
| KAA-CFGAT | 77.70 ±0.57 | 98.94 ±2.11 | 37.00 ±3.58 | **72.79** ±0.88 | 0.6726 ±0.0045 | 0.2398 ±0.0012 |
| GT | 94.50 ±0.36 | 89.47 ±3.33 | 50.33 ±3.23 | 72.43 ±1.22 | 0.5084 ±0.0190 | 0.1067 ±0.0128 |
| KAA-GT | **97.93** ±0.17 | 91.58 ±4.21 | **51.00** ±5.73 | 72.75 ±1.17 | 0.5042 ±0.0240 | **0.1056** ±0.0144 |
| SAN | 94.47 ±0.21 | 90.53 ±6.99 | 45.50 ±5.06 | 71.89 ±2.64 | 0.4935 ±0.0252 | 0.1145 ±0.0119 |
| KAA-SAN | 97.84 ±0.14 | 90.79 ±6.84 | 49.00 ±7.86 | 72.07 ±2.05 | **0.4675** ±0.0434 | 0.1076 ±0.0049 |
| **Avg Imp** $\uparrow$ (%) | +7.94 | +2.28 | +12.98 | +1.82 | +6.82 | +14.42 |

## 5.3 PERFORMANCE ON GRAPH-LEVEL TASKS

**Datasets Involved.** We conduct graph classification and graph regression tasks to validate the effectiveness of KAA in graph-level applications. For the graph classification task, we select four datasets (Ivanov et al., 2019; Zitnik & Leskovec, 2017) from bioinformatics and cheminformatics: *PPI, MUTAG, ENZYMES*, and *PROTEINS*. The downstream tasks for these datasets involve predicting the properties of proteins or molecules. For the graph regression tasks, we select two datasets: *ZINC* (Gómez-Bombarelli et al., 2018) and *QM9* (Wu et al., 2018). Both ZINC and QM9 are molecular datasets. The task for ZINC is to predict the constrained solubility of molecules, while QM9 involves regression tasks for 19 different molecular properties. More details and statistical information about these datasets can be found in Appendix B.1. We also employ GCN (Kipf & Welling, 2017), GraphSAGE (Hamilton et al., 2017), and GIN (Xu et al., 2019) as baseline models.

**Experimental Results.** The results of KAA on graph-level tasks are shown in Table 3. The experimental findings indicate that KAA-enhanced models consistently outperform the original models across all datasets. Notably, the performance improvement of KAA in graph-level tasks is more pronounced than in node-level tasks. Specifically, KAA achieves an average performance enhancement of 7.71% across all graph-level tasks, with over 10% improvement observed in one-third of the tasks and more than 5% improvement in two-thirds of the tasks. This remarkable performance underscores the effectiveness of KAA in graph-level tasks. For GAT-based models, we observe substantial performance gains in certain datasets with KAA, such as the enhancement of GLCN on the PPI dataset and GAT on the QM9 dataset. In these instances, KAA results in a qualitative leap in the performance of attentive GNN models, achieving improvements exceeding 20%. For Transformer-based models, which generally perform better on graph-level tasks, KAA still yields an average performance improvement of 2.72%. Overall, the outstanding performance of KAA in graph-level tasks further demonstrates the significant enhancement in model performance achieved by incorporating KAN into the scoring function.

## 6 CONCLUSION AND OUTLOOK

In our paper, we introduce Kolmogorov-Arnold Attention (KAA), which integrates KAN into the scoring functions of existing attentive GNN models. Through thorough theoretical and experimental validation, we demonstrate that KAN is highly effective for the scoring process, significantly enhancing the performance of KAA. Furthermore, our successful implementation of KAN in attentive GNNs may inspire advancements in KAN in other domains. Our comparative analysis of the theoretical expressive power of KAN-based learners versus MLP-based learners under constrained parameters offers valuable insights for future research.

## ACKNOWLEDGEMENTS

This work is supported by NSFC (No. 62322606), Zhejiang NSF (LR22F020005), and CCF-Zhipu Large Model Fund.

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

# A   EXTRA MATERIALS FOR SECTION 4

## A.1   CONNECTION BETWEEN KAA AND EXISTING ATTENTIVE TECHNIQUES

In fact, KAA is not at odds with advanced attentive GNN techniques and does not lose effectiveness due to their strong performance. This is because existing methods focus more on determining what constitutes important attention information, and advanced methods often excel at capturing such crucial information through the attention mechanism. KAA, on the other hand, enhances the actual construction process of the attention mechanism. This means that regardless of the type of attention distribution being learned, KAA can improve the success rate of accurately modeling that distribution. Therefore, despite the significant advancements in existing attentive GNNs, KAA can still provide performance improvements, as it represents an enhancement from a different perspective.

## A.2   DETAILS OF EXISTING MEASUREMENT

A pioneering work (Brody et al., 2021) defines static and dynamic attention. Here, we excerpt their formal definitions from the original text to understand the differences between static and dynamic attention and our defined MRD.

**(Static Attention)**   A (possibly infinite) family of scoring functions $\mathcal{F} \subseteq \left( \mathbb{R}^d \times \mathbb{R}^d \to \mathbb{R} \right)$ computes *static scoring* for a given set of key vectors $\mathbb{K} = \{ \boldsymbol{k}_1, ..., \boldsymbol{k}_n \} \subset \mathbb{R}^d$ and query vectors $\mathbb{Q} = \{ \boldsymbol{q}_1, ..., \boldsymbol{q}_m \} \subset \mathbb{R}^d$, if for every $f \in \mathcal{F}$ there exists a "highest scoring" key $j_f \in [n]$ such that for every query $i \in [m]$ and key $j \in [n]$ it holds that $f \left( \boldsymbol{q}_i, \boldsymbol{k}_{j_f} \right) \geq f \left( \boldsymbol{q}_i, \boldsymbol{k}_j \right)$. We say that a family of attention functions computes *static attention* given $\mathbb{K}$ and $\mathbb{Q}$, if its scoring function computes static scoring, possibly followed by monotonic normalization such as softmax.

**(Dynamic Attention)**   A (possibly infinite) family of scoring functions $\mathcal{F} \subseteq \left( \mathbb{R}^d \times \mathbb{R}^d \to \mathbb{R} \right)$ computes *dynamic scoring* for a given set of key vectors $\mathbb{K} = \{ \boldsymbol{k}_1, ..., \boldsymbol{k}_n \} \subset \mathbb{R}^d$ and query vectors $\mathbb{Q} = \{ \boldsymbol{q}_1, ..., \boldsymbol{q}_m \} \subset \mathbb{R}^d$, if for *any* mapping $\varphi \colon [m] \to [n]$ there exists $f \in \mathcal{F}$ such that for any query $i \in [m]$ and any key $j_{\neq \varphi(i)} \in [n]$: $f \left( \boldsymbol{q}_i, \boldsymbol{k}_{\varphi(i)} \right) > f \left( \boldsymbol{q}_i, \boldsymbol{k}_j \right)$. We say that a family of attention functions computes *dynamic attention* for $\mathbb{K}$ and $\mathbb{Q}$, if its scoring function computes dynamic scoring, possibly followed by monotonic normalization such as softmax.

## A.3   TRANSFORMING GAT INTO NON-STATIC ATTENTION

According to Formula 10, we add a negative sign and an absolute value function to the original scoring function of GAT, which can be expanded as follows:

$$\mathrm{s}(h_i, h_j) = -\mathrm{LeakyReLU}(\mathrm{Abs}(a^\top \cdot [\mathbf{W} h_i \| \mathbf{W} h_j])) \tag{18}$$

$$= -\mathrm{LeakyReLU}(|a_{:d'}^\top \cdot \mathbf{W} h_i + a_{d':}^\top \cdot \mathbf{W} h_j|) \tag{19}$$

$$= -\mathrm{LeakyReLU}(|b_1^\top \cdot h_i + b_2^\top \cdot h_j|) \tag{20}$$

where $\mathbf{W} \in \mathbb{R}^{d' \times d}$, $a \in \mathbb{R}^{2d' \times 1}$, and $a_{:d'}$ represents the vector composed of the first $d'$ dimensions, while $a_{d':}$ represents the vector composed of the last $d'$ dimensions. In Formula 20, we have $b_1^\top = a_{:d'}^\top \cdot \mathbf{W}$ and $b_2^\top = a_{d':}^\top \cdot \mathbf{W}$. Here, we can consider $b_1, b_2 \in \mathbb{R}^{d \times 1}$ as two free, learnable linear transformations. For a specific central node $h_i$, neighbor node $h_j$ with higher score needs to make $|b_1^\top \cdot h_i + b_2^\top \cdot h_j|$ as close to zero as possible, rather than maximizing it. Therefore, for different center nodes (queries) $h_i$, there theoretically exist different neighbor nodes (keys) $h_i$ that can achieve the highest scores. It conflicts with the definition of static attention. Thus, the scoring function of the form in Formula 10 belongs to non-static attention.

## A.4   ASSUMPTIONS AND SETUP FOR THEORETICAL ANALYSIS

In this section, we elaborate on the theoretical setup and assumptions, as well as some simplifications. Without loss of generality, we assume that the selected central node is connected to all other nodes in the graph (as most graph transformers compute the relationships between all nodes in this way). Additionally, for clarity in notation, we assume that $\mathrm{AF}(h_i, h_j) \in \mathbb{R}^d$, and denote the alignment matrix composed of all $\mathrm{AF}(h_i, h_j) \in \mathbb{R}^d$ values as $\mathbf{P} \in \mathbb{R}^{N \times d}$, where $N$ is the number of

involved nodes. In fact, the composition of the P matrix has a significant impact on subsequent analysis. For example, if $\mathbf{P} = \mathbf{1}^{N \times d}$, then regardless of the form of the scoring function, all nodes will receive the same score. We can see that if two nodes $j_1$ and $j_2$ have the same representations in $\mathbf{P}$ (*i.e.*, $\mathbf{P}_{j_1} = \mathbf{P}_{j_2}$), they are bound to receive the same score. Therefore, the distinguishability of $\mathbf{P}$ is crucial. To define a sufficiently distinguishable $\mathbf{P}$, we first provide the circulant matrix $\mathbf{C} \in \mathbb{R}^{N \times N}$, which can be expressed as:

$$\mathbf{C} = \begin{pmatrix} 1 & 2 & 3 & \cdots & N-1 & N \\ 2 & 3 & 4 & \cdots & N & 1 \\ 3 & 4 & 5 & \cdots & 1 & 2 \\ \vdots & \vdots & \vdots & \ddots & \vdots & \vdots \\ N-1 & N & 1 & \cdots & N-3 & N-2 \\ N & 1 & 2 & \cdots & N-2 & N-1 \end{pmatrix} \in \mathbb{R}^{N \times N} \tag{21}$$

$\mathbf{C}$ is a circulant matrix, so $\mathbf{C}$ is full-rank (Kra & Simanca, 2012). Additionally, $\mathbf{C}$ is sufficiently distinguishable. Specifically, the elements in each column of $\mathbf{C}$ are all different, and the arrangement of the relationships between the elements in different columns is also unique. Ideally, if $\mathbf{P} = \mathbf{C}$, the scoring function will have sufficient representations to generate a variety of rankings. In our assumption, the width of $\mathbf{P}$ is $d$, which is much smaller than $N$. For analytical simplification, we assume that $\mathbf{P}$ consists of the first $d$ columns of $\mathbf{C}$, and has the following form:

$$\mathbf{P} = \begin{pmatrix} 1 & 2 & 3 & \cdots & d-1 & d \\ 2 & 3 & 4 & \cdots & d & d+1 \\ 3 & 4 & 5 & \cdots & d+1 & d+2 \\ \vdots & \vdots & \vdots & \ddots & \vdots & \vdots \\ N-1 & N & 1 & \cdots & d-3 & d-2 \\ N & 1 & 2 & \cdots & d-2 & d-1 \end{pmatrix} \in \mathbb{R}^{N \times d} \tag{22}$$

Such a $\mathbf{P}$ will serve as the input for the respective scoring functions discussed later. In addition, we make an assumption regarding the magnitudes of $N$ and $d$: $N$ is much greater than $d$ ($N \gg d$), and we assume $N = d^2$. Finally, we assume that to obtain an importance ranking $\sigma$, the scoring function $\mathrm{s}(\cdot)$ must satisfy:

$$\forall j \in [1, N], \quad \mathrm{s}(\mathbf{P}_j) = \sigma^{-1}(j) \tag{23}$$

Here, we use $\mathrm{s}(\mathbf{P}_j)$ to replace $\mathrm{s}(h_i, h_j)$, as they convey the same meaning. Formula 23 establishes the connection between the score and the ranking, which is natural and fair. Based on this requirement, the MRD in Formula 13 can be specifically calculated as follows:

$$\mathrm{MRD}(\mathcal{S}, \mathbf{P}) = \max_{\pi' \in \Pi} \min_{s' \in \mathcal{S}} \sqrt{\sum_{j}^{N} (\mathrm{s}'(\mathbf{P}_j) - \pi'^{-1}(j))^2} \tag{24}$$

where $\mathcal{S}$ is the set of all candidate scoring functions, and $\Pi$ is the set of all permutations of $N$. The MRD of various scoring functions discussed later will all be calculated as Formula 24.

## A.5 DETAILS FOR MRD OF LINEAR TRANSFORMATION-BASED ATTENTION

In this section, we provide a detailed proof of Proposition 1.

**Proposition 1** (MRD of Linear Transformation-Based Attention). *Given the alignment matirx* $\mathbf{P} \in \mathbb{R}^{N \times d}$, *for a scoring function in the form of* $\mathrm{s}(h_i, h_j) = \mathbf{W} \cdot \mathrm{AF}(h_i, h_j)$, *where* $\mathbf{W} \in \mathbb{R}^{d \times 1}$, *its MRD satisfies the following inequality:*

$$\mathrm{MRD}(\mathcal{S}_{\mathrm{LT}}, \mathbf{P}) \geq \sqrt{\frac{1}{12}(N^3 - N - d^3 + 3d^2 - 2d)}$$

*where* $\mathcal{S}_{\mathrm{LT}}$ *is the set of all candidate linear transformation-based scoring functions.*

*Proof.* For linear transformation-based scoring functions, they can linearly combine the column vectors of $\mathbf{P}$ in Formula 22 to obtain the final scores. Therefore, we need to consider the range

space of the linear combinations of $\mathbf{P}$. In this case, a series of column operations on $\mathbf{P}$ will not affect the final result. First, starting from the second column of $\mathbf{P}$, we subtract the values of the previous column from each column to obtain $\mathbf{P}'$ as follows:

$$
\mathbf{P}' = \begin{pmatrix}
1 & 1 & 1 & \cdots & 1 \\
2 & 1 & 1 & \cdots & \vdots \\
3 & 1 & 1 & \cdots & 1-N \\
\vdots & \vdots & \vdots & \ddots & \vdots \\
N-1 & 1 & 1-N & \cdots & 1 \\
N & 1-N & 1 & \cdots & 1
\end{pmatrix} \in \mathbb{R}^{N \times d}
\tag{25}
$$

In $\mathbf{P}'$, except for the first column, which remains $\mathbf{P}'_{:,1} = [1, 2, 3, ..., N]^\top$, the other column vectors consist of $(N-1)$ ones and an $(1-N)$ that gradually changes position. Then, starting from the third column of $\mathbf{P}'$, we again subtract the values of the previous column from each column and divide the resulting vectors by $N$ to obtain $\mathbf{P}^*$ as follows:

$$
\mathbf{P}^* = \begin{pmatrix}
1 & 1 & 0 & 0 & \cdots & 0 \\
2 & 1 & 0 & 0 & \cdots & 0 \\
3 & 1 & 0 & 0 & \cdots & 0 \\
4 & 1 & 0 & 0 & \cdots & 0 \\
\vdots & \vdots & \vdots & \vdots & \ddots & \vdots \\
N-3 & 1 & 0 & 0 & \cdots & -1 \\
N-2 & 1 & 0 & -1 & \cdots & 1 \\
N-1 & 1 & -1 & 1 & \cdots & \vdots \\
N & 1-N & 1 & 0 & \cdots & 0
\end{pmatrix} \in \mathbb{R}^{N \times d}
\tag{26}
$$

The elements in $\mathbf{P}^*$ are composed in a regular pattern, allowing it to be divided into two parts. The upper part consists of the first $(N + 1 - d)$ rows, denoted as $\mathbf{P}^*_{:(N+1-d)}$, as shown below:

$$
\mathbf{P}^*_{:(N+1-d)} = \begin{pmatrix}
1 & 1 & 0 & 0 & \cdots & 0 \\
2 & 1 & 0 & 0 & \cdots & 0 \\
3 & 1 & 0 & 0 & \cdots & 0 \\
4 & 1 & 0 & 0 & \cdots & 0 \\
\vdots & \vdots & \vdots & \vdots & \ddots & \vdots \\
N+1-d & 1 & 0 & 0 & \cdots & 0
\end{pmatrix} \in \mathbb{R}^{(N+1-d) \times d}
\tag{27}
$$

The scores of the first $(N + 1 - d)$ nodes will be formed by linear combinations of the column vectors of $\mathbf{P}^*_{:(N+1-d)}$. The lower part of $\mathbf{P}^*$ consists of the last $(d-1)$ rows, denoted as $\mathbf{P}^*_{(N+1-d):}$, as shown below:

$$
\mathbf{P}^*_{(N+1-d):} = \begin{pmatrix}
N+2-d & 1 & 0 & 0 & \cdots & -1 \\
N+3-d & 1 & 0 & 0 & \cdots & 1 \\
\vdots & \vdots & \vdots & \vdots & \ddots & \vdots \\
N-2 & 1 & 0 & -1 & \cdots & 0 \\
N-1 & 1 & -1 & 1 & \cdots & 0 \\
N & 1-N & 1 & 0 & \cdots & 0
\end{pmatrix} \in \mathbb{R}^{(d-1) \times d}
\tag{28}
$$

The scores of the last $(d - 1)$ nodes will be formed by linear combinations of the column vectors of $\mathbf{P}^*_{(N+1-d):}$. According to Formula 24, the MRD can be split into two parts for calculation, as shown below:

$$
\mathrm{MRD}(\mathcal{S}_{\mathrm{LT}}, \mathbf{P}) = \max_{\pi' \in \Pi} \min_{s' \in \mathcal{S}_{\mathrm{LT}}} \sqrt{\sum_{j}^{N} (s'(\mathbf{P}_j) - \pi'^{-1}(j))^2}
\tag{29}
$$

$$
= \max_{\pi' \in \Pi} \min_{s' \in \mathcal{S}_{\mathrm{LT}}} \sqrt{\sum_{i=1}^{N+1-d} (s'(\mathbf{P}_i) - \pi'^{-1}(i))^2 + \sum_{j=N+2-d}^{N} (s'(\mathbf{P}_j) - \pi'^{-1}(j))^2}
\tag{30}
$$

In Formula 30, the calculation of the entire MRD is divided into two parts. The first part includes the scores of the first $(N + 1 - d)$ nodes determined by $\mathbf{P}^*_{:(N+1-d)}$, and the second part includes the scores of the last $(d-1)$ nodes determined by $\mathbf{P}^*_{(N+1-d):}$. According to Formula 27, the scores of the first $(N+1-d)$ nodes are composed of only two non-zero column vectors, $[1, 2, 3, ..., N+1-d]^\top \in \mathbb{R}^{(N+1-d)\times 1}$ and $\mathbf{1} \in \mathbb{R}^{(N+1-d)\times 1}$. Therefore, the score $s(\mathbf{P}_i)$ satisfies the following equation:

$$\forall i \in [1, N + 1 - d], s(\mathbf{P}_i) = ai + b \tag{31}$$

where $a, b \in \mathbb{R}$ are two learnable parameters. According to Formula 28, the column rank of $\mathbf{P}^*_{(N+1-d):}$ is $(d-1)$, which means that $\mathbf{P}^*_{(N+1-d):}$ is a full column-rank matrix. Therefore, Formula 30 can be scaled as follows:

$$\text{MRD}(\mathcal{S}_{\text{LT}}, \mathbf{P}) = \max_{\pi' \in \Pi} \min_{s' \in \mathcal{S}_{\text{LT}}} \sqrt{\sum_{i=1}^{N+1-d} (s'(\mathbf{P}_i) - \pi'^{-1}(i))^2 + \sum_{j=N+2-d}^{N} (s'(\mathbf{P}_j) - \pi'^{-1}(j))^2} \tag{32}$$

$$\geq \min_{s' \in \mathcal{S}_{\text{LT}}} \sqrt{\max_{\pi' \in \Pi} \sum_{i=1}^{N+1-d} (s'(\mathbf{P}_i) - \pi'^{-1}(i))^2 + \sum_{j=N+2-d}^{N} (s'(\mathbf{P}_j) - \pi'^{-1}(j))^2} \tag{33}$$

$$\geq \min_{s' \in \mathcal{S}_{\text{LT}}} \sqrt{\max_{\pi' \in \Pi} \sum_{i=1}^{N+1-d} (s'(\mathbf{P}_i) - \pi'^{-1}(i))^2} \tag{34}$$

$$= \min_{a,b \in \mathbb{R}} \sqrt{\max_{\pi' \in \Pi} \sum_{i=1}^{N+1-d} (ai + b - \pi'^{-1}(i))^2} \tag{35}$$

To maximize the term under the square root in Formula 35, the first $(N + 1 - d)$ elements of $\pi'$ should be uniformly distributed (i.e., $\pi'^{-1}(1) = 1, \pi'^{-1}(2) = N, \pi'^{-1}(3) = 2, \pi'^{-1}(4) = N - 1, ...$) (Davis, 1979). In this case, the minimum value is obtained when $a = 0$ and $b = \frac{N+1}{2}$. Therefore, $\text{MRD}(\mathcal{S}_{\text{LT}}, \mathbf{P})$ has the following lower bound:

$$\text{MRD}(\mathcal{S}_{\text{LT}}, \mathbf{P}) \geq \min_{a,b \in \mathbb{R}} \sqrt{\max_{\pi' \in \Pi} \sum_{i=1}^{N+1-d} (ai + b - \pi'^{-1}(i))^2} \tag{36}$$

$$= \sqrt{2((N - \frac{N+1}{2})^2 + (N - 1 - \frac{N+1}{2})^2 + ... + (\frac{d}{2})^2)} \tag{37}$$

$$= \sqrt{\frac{1}{12}(N^3 - N - d^3 + 3d^2 - 2d)} \tag{38}$$

Now, we have completed the derivation of Proposition 1. $\qquad\square$

## A.6 DETAILS FOR MRD OF MLP-BASED ATTENTION

In this section, we provide a detailed proof of Proposition 2.

**Proposition 2** (MRD of MLP-Based Attention). *Given the alignment matirx* $\mathbf{P} \in \mathbb{R}^{N \times d}$, *for a scoring function in the form of* $s(h_i, h_j) = \mathbf{W}_2 \cdot (\text{ReLU}(\mathbf{W}_1 \cdot \text{AF}(h_i, h_j)))$, *where* $\mathbf{W}_1 \in \mathbb{R}^{d \times d}$ *and* $\mathbf{W}_2 \in \mathbb{R}^{d \times 1}$, *its MRD satisfies the following inequality:*

$$\sqrt{\frac{1}{12}(N^3 - N - \lambda)} \geq \text{MRD}(\mathcal{S}_{\text{MLP}}, \mathbf{P}) \geq \sqrt{\frac{1}{12}((N-d)^3 - (N-d) - \lambda)}$$

*where* $\lambda = d^3 - 3d^2 + 2d$ *and* $\mathcal{S}_{\text{MLP}}$ *is the set of all candidate MLP-based scoring functions.*

*Proof.* We decompose the scoring process of the two-layer MLP-based scoring function into three parts: the first linear transformation $\mathbf{W}_1$, the nonlinear activation $\text{ReLU}(\cdot)$, and the second linear transformation $\mathbf{W}_2$. First, the initial linear transformation $\mathbf{W}_1$ transform $\mathbf{P}$. As in Appendix A.5, we can also convert $\mathbf{P}$ from Formula 22 into the more analytically convenient form $\mathbf{P}^*$ as Formula

26. Additionally, $\mathbf{P}^*$ can still be divided into the upper part $\mathbf{P}^*_{:(N+1-d)}$ as in Formula 27 and the lower part $\mathbf{P}^*_{(N+1-d):}$ as in Formula 28. Next, we consider the impact of ReLU($\cdot$) on the obtained scores. Specifically, ReLU($\cdot$) is defined as follows:

$$\text{ReLU}(x) = \max(0, x) \tag{39}$$

The linear transformation produces a linear combination of the original column vectors, but ReLU($\cdot$) will generate a series of new column vectors. These newly generated column vectors serve as the basis for the second linear transformation, which then performs another linear combination. $\mathbf{P}^*_{:(N+1-d)}$ contains only two non-zero column vectors, and their linear combination, after applying ReLU($\cdot$), results in the following set of column vectors $\boldsymbol{c}$:

$$\boldsymbol{c} = \text{ReLU}(a \cdot [1, 2, 3, ..., N+1-d]^\top + b \cdot \mathbf{1}) \in \mathcal{C} \tag{40}$$

$$\mathcal{C} = \{\boldsymbol{c} |\ \exists\ 1 \leq i \leq N+1-d, \boldsymbol{c} = [a+b, 2a+b, ..., ia+b, 0, ..., 0]^\top$$

$$\text{or } \boldsymbol{c} = [0, ..., 0, ia+b, (i+1)a+b, ..., (N+1-d)a+b]^\top\} \tag{41}$$

Next, we analyze the column vectors of $\mathbf{P}^*_{(N+1-d):}$ after the linear transformation and ReLU($\cdot$). Here, we focus on the last $(d-2)$ columns of $\mathbf{P}^*_{(N+1-d):}$, which can be expressed as:

$$\mathbf{P}^*_{(N+1-d):,2:} = \begin{pmatrix} 0 & 0 & \cdots & -1 \\ 0 & 0 & \cdots & 1 \\ \vdots & \vdots & \ddots & \vdots \\ 0 & -1 & \cdots & 0 \\ -1 & 1 & \cdots & 0 \\ 1 & 0 & \cdots & 0 \end{pmatrix} \in \mathbb{R}^{(d-1) \times (d-2)} \tag{42}$$

By applying ReLU($\cdot$) and the linear transformation, we obtain the following matrix $\mathbf{M}$:

$$\mathbf{M} = [\text{ReLU}(\mathbf{P}^*_{(N+1-d):,2:}) | \text{ReLU}(-\mathbf{P}^*_{(N+1-d):,d})] \tag{43}$$

$$= \begin{pmatrix} 0 & 0 & \cdots & 0 & 1 \\ 0 & 0 & \cdots & 1 & 0 \\ \vdots & \vdots & \ddots & \vdots & \vdots \\ 0 & 0 & \cdots & 0 & 0 \\ 0 & 1 & \cdots & 0 & 0 \\ 1 & 0 & \cdots & 0 & 0 \end{pmatrix} \in \mathbb{R}^{(d-1) \times (d-1)} \tag{44}$$

$\mathbf{M}$ has ones on one diagonal, while all other elements are zero. Clearly, $\mathbf{M}$ is of full rank. Next, we will begin calculating the upper and lower bounds of $\text{MRD}(\mathcal{S}_{\text{MLP}}, \mathbf{P})$. First, we consider its upper bound. We assume that after the first linear transformation $\mathbf{W}_1$ and ReLU($\cdot$), we obtain the following intermediate representations $\mathbf{H}$:

$$\mathbf{H} = \begin{pmatrix} \mathbf{1} & \mathbf{0} \\ 1-N & \mathbf{M} \end{pmatrix} \in \mathbb{R}^{N \times d} \tag{45}$$

$\mathbf{H}$ is composed of the second column from $\mathbf{P}^*$ and $\mathbf{M}$, and it can be obtained through the linear transformation and ReLU($\cdot$) according to Formula 43. Thus, $\text{MRD}(\mathcal{S}_{\text{MLP}}, \mathbf{P})$ is transformed into $\text{MRD}(\mathcal{S}_{\text{LT}}, \mathbf{H})$, which can be calculated as follows:

$$\text{MRD}(\mathcal{S}_{\text{MLP}}, \mathbf{P}) \leq \text{MRD}(\mathcal{S}_{\text{LT}}, \mathbf{H}) \tag{46}$$

$$= \max_{\pi' \in \Pi} \min_{s' \in \mathcal{S}_{\text{LT}}} \sqrt{\sum_{i=1}^{N+1-d} (s'(\mathbf{H}_i) - \pi'^{-1}(i))^2 + \sum_{j=N+2-d}^{N} (s'(\mathbf{H}_j) - \pi'^{-1}(j))^2} \tag{47}$$

$$= \max_{\pi' \in \Pi} \min_{a \in \mathbb{R}} \sqrt{\sum_{i=1}^{N+1-d} a - \pi'^{-1}(i))^2} \tag{48}$$

In this case, the scores for the first $(N+1-d)$ nodes are all the same, while the scores for the last $(d-1)$ nodes can take on any value due to the full rank of $\mathbf{M}$. Therefore, to compute Formula 48,

the first $(N+1-d)$ elements of $\pi'$ should be uniformly distributed (*i.e.*, $\pi'^{-1}(1) = 1, \pi'^{-1}(2) = N, \pi'^{-1}(3) = 2, \pi'^{-1}(4) = N - 1, ...$) (Davis, 1979). In that case, $a$ should be $\frac{n+1}{2}$. Thus, we can obtain the upper bound of $\text{MRD}(\mathcal{S}_{\text{MLP}}, \mathbf{P})$:

$$\text{MRD}(\mathcal{S}_{\text{MLP}}, \mathbf{P}) \leq \max_{\pi' \in \Pi} \min_{a \in \mathbb{R}} \sqrt{\sum_{i=1}^{N+1-d} a - \pi'^{-1}(i))^2} \tag{49}$$

$$= \sqrt{2((N - \frac{N+1}{2})^2 + (N - 1 - \frac{N+1}{2})^2 + ... + (\frac{d}{2})^2)} \tag{50}$$

$$= \sqrt{\frac{1}{12}(N^3 - N - d^3 + 3d^2 - 2d)} \tag{51}$$

Next, we calculate the lower bound of $\text{MRD}(\mathcal{S}_{\text{MLP}}, \mathbf{P})$. Similar to Formula 30, we have:

$$\text{MRD}(\mathcal{S}_{\text{MLP}}, \mathbf{P}) = \max_{\pi' \in \Pi} \min_{s' \in \mathcal{S}_{\text{MLP}}} \sqrt{\sum_{j}^{N} (s'(\mathbf{P}_j) - \pi'^{-1}(j))^2} \tag{52}$$

$$= \max_{\pi' \in \Pi} \min_{s' \in \mathcal{S}_{\text{MLP}}} \sqrt{\sum_{i=1}^{N+1-d} (s'(\mathbf{P}_i) - \pi'^{-1}(i))^2 + \sum_{j=N+2-d}^{N} (s'(\mathbf{P}_j) - \pi'^{-1}(j))^2} \tag{53}$$

$$\geq \max_{\pi' \in \Pi} \min_{s' \in \mathcal{S}_{\text{MLP}}} \sqrt{\sum_{i=1}^{N+1-d} (s'(\mathbf{P}_i) - \pi'^{-1}(i))^2} \tag{54}$$

To compute the lower bound of Formula 54, the intermediate representations $\mathbf{H}'$ should be entirely composed of the column vectors from the set $\mathcal{C}$ in Formula 41. Since the intermediate layer dimension of the two-layer MLP is $d$, we can select at most $d$ different column vectors to include in $\mathbf{H}'$. To obtain the lower bound of $\text{MRD}(\mathcal{S}_{\text{MLP}})$, we expand the intermediate representations $\mathbf{H}'$ to retain the original column vectors $[1, 2, 3, ..., N+1-d]^\top \in \mathbb{R}^{(N+1-d) \times 1}$ and $\mathbf{1} \in \mathbb{R}^{(N+1-d) \times 1}$ from $\mathbf{P}^*_{:(N+1-d)}$, while additionally selecting $d$ column vectors from $\mathcal{C}$ that have all their zero values at the beginning. This simplification does not affect the results because the column vectors with zero values at the end can be derived from $[1, 2, 3, ..., N+1-d]^\top$ by subtracting the vectors with zero values at the beginning, thus not affecting the range space. We denote the expanded intermediate representations as $\mathbf{H}^* \in \mathbb{R}^{(N+1-d) \times (d+2)}$, and denote its candidate set as $\mathcal{H}$:

$$\mathbf{H}^* = \begin{pmatrix} 1 & 1 \\ 2 & 1 \\ \vdots & \vdots & \boldsymbol{c}_1 & \boldsymbol{c}_2 & \cdots & \boldsymbol{c}_d \\ N+1-d & 1 \end{pmatrix} \in \mathcal{H} \tag{55}$$

Thus, we can further expand Formula 54 as follows:

$$\text{MRD}(\mathcal{S}_{\text{MLP}}, \mathbf{P}) \geq \max_{\pi' \in \Pi} \min_{s' \in \mathcal{S}_{\text{MLP}}} \sqrt{\sum_{i=1}^{N+1-d} (s'(\mathbf{P}_i) - \pi'^{-1}(i))^2} \tag{56}$$

$$\geq \max_{\pi' \in \Pi} \min_{s' \in \mathcal{S}_{\text{LT}}} \min_{\mathbf{H}^* \in \mathcal{H}} \sqrt{\sum_{i=1}^{N+1-d} (s'(\mathbf{H}^*_i) - \pi'^{-1}(i))^2} \tag{57}$$

According to Formula 57, we have transformed the MRD of the MLP back into the MRD of a linear transformation. To reach the extremum, the first $(N+1-d)$ elements of $\pi'$ should also be uniformly distributed (*i.e.*, $\pi'^{-1}(1) = 1, \pi'^{-1}(2) = N, \pi'^{-1}(3) = 2, \pi'^{-1}(4) = N-1, ...$) (Davis, 1979). In this case, any $\boldsymbol{c}_i$ (including $[1, 2, 3, ..., N+1-d]^\top$) is less effective than simply retaining the column vector $\boldsymbol{z}_i$ with its first non-zero element:

$$\boldsymbol{c}_i = [0, ..., 0, ja + b, (j+1)a + b, ..., (N+1-d)a + b]^\top, 1 \leq j \leq N+1-d \tag{58}$$

$$\boldsymbol{z}_i = [0, ..., 0, ja + b, 0, ..., 0]^\top, 1 \leq j \leq N+1-d \tag{59}$$

Therefore, by replacing all $c_i$ with $z_i$ (replacing $[1, 2, 3, ..., N+1-d]^\top$ with $z_0$), we construct new intermediate representations $\mathbf{Z}^* \in \mathbb{R}^{(N+1-d)\times(d+2)}$, and its candidate set is denoted as $\mathcal{Z}$:

$$\mathbf{Z}^* = \begin{pmatrix} 1 \\ 1 \\ \vdots & z_0 & z_1 & z_2 & \cdots & z_d \\ 1 \end{pmatrix} \in \mathcal{Z} \tag{60}$$

Then, we continue to calculate the lower bound of $\mathrm{MRD}(\mathcal{S}_{\mathrm{MLP}}, \mathbf{P})$:

$$\mathrm{MRD}(\mathcal{S}_{\mathrm{MLP}}, \mathbf{P}) \geq \max_{\pi' \in \Pi} \min_{s' \in \mathcal{S}_{\mathrm{LT}}} \min_{\mathbf{H}^* \in \mathcal{H}} \sqrt{\sum_{i=1}^{N+1-d} (\mathrm{s}'(\mathbf{H}_i^*) - \pi'^{-1}(i))^2} \tag{61}$$

$$= \min_{s' \in \mathcal{S}_{\mathrm{LT}}} \min_{\mathbf{H}^* \in \mathcal{H}} \sqrt{\sum_{i=1}^{N+1-d} (\mathrm{s}'(\mathbf{H}_i^*) - \pi'^{-1}(i))^2} \tag{62}$$

$$\geq \min_{s' \in \mathcal{S}_{\mathrm{LT}}} \min_{\mathbf{Z}^* \in \mathcal{Z}} \sqrt{\sum_{i=1}^{N+1-d} (\mathrm{s}'(\mathbf{Z}_i^*) - \pi'^{-1}(i))^2} \tag{63}$$

According to Formula 60, each $z_i$ can provide an accurate score for a certain node. The $(d+1)$ column vectors from $z_0$ to $z_d$ should eliminate errors for the $(d+1)$ points that are farthest from the mean, in order to achieve global optimality. Therefore, we have:

$$\mathrm{MRD}(\mathcal{S}_{\mathrm{MLP}}, \mathbf{P}) \geq \min_{s' \in \mathcal{S}_{\mathrm{LT}}} \min_{\mathbf{Z}^* \in \mathcal{Z}} \sqrt{\sum_{i=1}^{N+1-d} (\mathrm{s}'(\mathbf{Z}_i^*) - \pi'^{-1}(i))^2} \tag{64}$$

$$\geq \min_{a \in \mathbb{R}} \sqrt{\sum_{i=1+\frac{d+1}{2}}^{N+1-d-\frac{d+1}{2}} (a - \pi'^{-1}(i))^2} \tag{65}$$

$$= \sqrt{\frac{1}{12}((N-d)^3 - (N-d) - d^3 + 3d^2 - 2d)} \tag{66}$$

Now, we have derived both the upper and lower bounds for $\mathrm{MRD}(\mathcal{S}_{\mathrm{MLP}}, \mathbf{P})$ and completed the derivation of Proposition 2. $\qquad\square$

### A.7 DETAILS FOR MRD OF KOLMOGOROV-ARNOLD ATTENTION

In this section, we provide a detailed proof of Proposition 3.

**Proposition 3** (MRD of Kolmogorov-Arnold Attention). *Given the alignment matirx $\mathbf{P} \in \mathbb{R}^{N \times d}$, for a scoring function in the form of $\mathrm{s}(h_i, h_j) = \sum_{k=1}^{d} \phi_k(\mathrm{AF}(h_i, h_j)_k)$, where each $\phi_k$ is composed of $d$ modified zero-order B-spline functions $\phi_k(x) = \sum_{l=1}^{d} c_{k,l} \cdot B_{k,l}^*(x)$, its MRD satisfies the following inequality:*

$$\mathrm{MRD}(\mathcal{S}_{\mathrm{KAA}}, \mathbf{P}) \leq \delta, \quad for \; \forall \, \delta > 0 \tag{67}$$

*where $\mathcal{S}_{\mathrm{KAA}}$ is the set of all candidate KAA scoring functions.*

First, we present the specific expression for the modified zero-order B-spline functions $B^*(\cdot)$ as follows:

$$B_j^*(x) := \begin{cases} 0, & t_j < x \leq t_{j+1} - 1 \\ 1, & t_{j+1} - 1 < x \leq t_{j+1} \\ 0, & \text{otherwise} \end{cases} \tag{68}$$

The modified $B^*(\cdot)$ is not significantly different from the standard zero-order B-spline function, except that it reduces the domain where the function takes non-zero values. For each non-linear

mapping function $\phi(\cdot)$, it consists of $d$ modified $B^*(\cdot)$, as shown below:

$$\phi(x) = \sum_{l=1}^{d} c_l \cdot B_l^*(x) \tag{69}$$

In our assumption, we have $N = d^2$, and according to Formula 22, the elements in $\mathbf{P}$ take values from $[1, N]$. Therefore, the grid size of $B^*(\cdot)$ we used is also $d$, and its specific form is as follows:

$$\forall\, l \in [1, d],\ B_l^*(x) := \begin{cases} 0, & (l-1)d < x \le ld - 1 \\ 1, & ld - 1 < x \le ld \\ 0, & \text{otherwise} \end{cases} \tag{70}$$

Then, we can express the scores of $\mathbf{P}_j \in \mathbb{R}^{1 \times d}$ in the form consisting of $B^*(\cdot)$ and learnable parameters $c$ as follows:

$$s(\mathbf{P}_j) = \sum_{k=1}^{d} \phi_k(\mathbf{P}_{j,k}) \tag{71}$$

$$= \sum_{k=1}^{d} \sum_{l=1}^{d} c_{k,l} B_{k,l}^*(\mathbf{P}_{j,k}) \tag{72}$$

where $\mathbf{P}_{j,k} \in \mathbb{R}$ is the $k$-th dimension value of $\mathbf{P}_j$. According to Formula 70, we can observe that $B^*(\cdot)$ only produces a non-zero value for integer input $x \in N^*$ when $x$ is divisible by $d$. For any $\mathbf{P}_j$, its $d$-dimensional values are composed of $d$ consecutive integers, so there exists exactly one $k \in [1, d]$ that satisfies $d \mid \mathbf{P}_{j,k}$. Therefore, only one $B_k^*(\cdot)$ is activated. Specifically, we have:

$$\forall j \in N^*,\ \text{and}\ 1 \le j \le N,$$
$$\exists \alpha, \beta \in N, 0 \le \alpha, \beta \le d - 1,$$
$$\text{s.t.}\quad j = \alpha d + \beta + 1 \tag{73}$$

By converting the row index $j$ corresponding to $\mathbf{P}_j$ into the form in Formula 73, we can obtain the value of $s(\mathbf{P}_j)$ as:

$$s(\mathbf{P}_j) = \sum_{k=1}^{d} \sum_{l=1}^{d} c_{k,l} B_{k,l}^*(\mathbf{P}_{j,k}) \tag{74}$$

$$= c_{d-\beta, \alpha+1} \tag{75}$$

Therefore, $\text{MRD}(\mathcal{S}_{\text{KAA}}, \mathbf{P})$ can be calculated as follows:

$$\text{MRD}(\mathcal{S}_{\text{KAA}}, \mathbf{P}) = \max_{\pi' \in \Pi} \min_{s' \in \mathcal{S}_{\text{KAA}}} \sqrt{\sum_{j}^{N} (s'(\mathbf{P}_j) - \pi'^{-1}(j))^2} \tag{76}$$

$$= \max_{\pi' \in \Pi} \min_{s' \in \mathcal{S}_{\text{KAA}}} \sqrt{\sum_{\alpha=0}^{d-1} \sum_{\beta=0}^{d-1} (s'(\mathbf{P}_{\alpha d + \beta + 1}) - \pi'^{-1}(\alpha d + \beta + 1))^2} \tag{77}$$

$$= \max_{\pi' \in \Pi} \min_{s' \in \mathcal{S}_{\text{KAA}}} \sqrt{\sum_{\alpha=0}^{d-1} \sum_{\beta=0}^{d-1} (c_{d-\beta, \alpha+1} - \pi'^{-1}(\alpha d + \beta + 1))^2} \tag{78}$$

According to Formula 78, for any permutation $\pi \in \Pi$, we can find a group of $c_{d-\beta, \alpha+1} = \pi'^{-1}(\alpha d + \beta + 1)$ such that $\text{MRD}(\mathcal{S}_{\text{KAA}}, \mathbf{P})$ can be made arbitrarily small. Thus, Proposition 3 is proven.

## B  MORE INFORMATION ON EXPERIMENTS

### B.1  DETAILED DESCRIPTIONS OF DATASETS

**Node-level datasets**  We employ 6 datasets for our node-level tasks, which includes 4 citation network datasets *Cora, CiteSeer, PubMed, ogbn-arxiv* and 2 product network datasets *Amazon-Computers, Amazon-Photo*. The statistics of above datasets are presented in Table 4.

- *Cora, CiteSeer, PubMed, ogbn-arxiv:* These 4 widely used graph datasets (Sen et al., 2008; Hu et al., 2020) represent the citation network, where nodes correspond to research papers, and edges indicate citation relationships. The downstream tasks typically involve classifying the research area of papers/researchers, and predicting the citation relationships.

- *Amazon-Computers, Amazon-Photo:* These are 2 product networks (Shchur et al., 2018) from Amazon, where nodes represent goods and edges indicate that two goods are frequently purchased together. The downstream task is to categorize goods to their corresponding product category.

Table 4: Statistics of node-level datasets.

| Dataset | Nodes | Edges | Feature | Classes | Train/Val/Test |
|---|---|---|---|---|---|
| Cora | 2708 | 5429 | 1433 | 7 | 140 / 500 / 1000 |
| CiteSeer | 3327 | 4732 | 3703 | 6 | 120 / 500 / 1000 |
| PubMed | 19717 | 44338 | 500 | 3 | 60 / 500 / 1000 |
| ogbn-arxiv | 169343 | 1166243 | 128 | 40 | 90941 / 29799 / 48603 |
| Amazon-Computers | 13752 | 491722 | 767 | 10 | 10% / 10% / 80% |
| Amazon-Photo | 7650 | 238162 | 745 | 8 | 10% / 10% / 80% |

**Graph-level datasets** We employ another 6 graph datasets for our graph-level tasks, which includes 1 biological network dataset *PPI*, 2 chemical compound graph datasets *MUTAG, ZINC*, 2 protein structure datasets *PROTEINS, ENZYMES*, and 1 quantum chemistry dataset *QM9*. The statistics of above datasets are presented in Table 5.

- *PPI:* This dataset (Zitnik & Leskovec, 2017) is a collection of biomedical system graphs, where each node represents a protein, and each edge represents the protein-protein interaction. The downstream tasks often involves graph classification on the protein's properties.

- *MUTAG, ZINC:* These 2 datasets belong to chemical compound graphs, where nodes are atoms and edges are chemical bonds. MUTAG (Ivanov et al., 2019) focuses on classifying compounds as mutagenic or non-mutagenic, while ZINC (Gómez-Bombarelli et al., 2018) focuses on predicting molecule properties related to drug discovery.

- *PROTEINS, ENZYMES:* These are 2 protein structure datasets (Ivanov et al., 2019). The nodes here represent secondary structure elements in the protein and edges represent the interaction between them. PROTEINS is used to classify proteins as enzymes or non-enzymes, while ENZYMES categorizes the proteins into one of six enzyme types.

- *QM9:* This dataset (Wu et al., 2018) is a collection of small organic molecules. Nodes in the graph represent atoms, and edges represent chemical bonds. The dataset is used for predicting several quantum mechanical properties of a molecule.

Table 5: Statistics of graph-level datasets.

| Dataset | Graphs | Nodes *(avg)* | Edges *(avg)* | Feature | Classes | Train/Val/Test |
|---|---|---|---|---|---|---|
| PPI | 24 | 2414 | 33838 | 50 | 121 | 80% / 10% / 10% |
| MUTAG | 188 | 17.93 | 19.76 | 7 | 2 | 80% / 10% / 10% |
| ENZYMES | 600 | 32.82 | 62.60 | 3 | 6 | 80% / 10% / 10% |
| PROTEINS | 1113 | 39.72 | 74.04 | 3 | 2 | 80% / 10% / 10% |
| ZINC | 12000 | 23.15 | 24.91 | 1 | 1 | 10000 / 1000 / 1000 |
| QM9 | 130831 | 18.03 | 18.66 | 11 | 12 | 80% / 10% / 10% |

## B.2 COMPARISON WITH MLP-BASED ATTENTION

In this section, we compare the performance between our proposed KAA and the MLP-based scoring function. The MLP-based scoring function is formally defined and analyzed by Brody et al. (2021). They proposed GATv2, which introduces an MLP into the scoring function based on GAT. We compare the performance of GAT, GATv2, and KAA-GAT on four datasets: Cora, CiteSeer, PubMed,

and ogbn-arxiv, with the results shown in Table 6. The experimental results show that KAA-GAT achieves the best performance on all datasets. Furthermore, although GATv2 has a stronger theoretical expressive capability, it does not outperform GAT on half of the datasets. These results also demonstrate that our proposed KAA not only possesses stronger theoretical expressiveness but also exhibits better performance on downstream tasks.

Table 6: Results of accuracy (%) comparison between GAT, GATv2, and our proposed KAA-GAT.

| Dataset / Model | Cora | CiteSeer | PubMed | ogbn-arxiv |
|---|---|---|---|---|
| GAT | 82.76 ±0.74 | 72.34 ±0.44 | 77.50 ±0.75 | 68.62 ±0.68 |
| GATv2 | 82.95 ±0.69 | 72.03 ±0.66 | 78.14 ±0.38 | 68.31 ±0.18 |
| KAA-GAT | **83.80** ±0.49 | **73.10** ±0.61 | **78.60** ±0.50 | **69.02** ±0.49 |

### B.3 COMPARISON WITH OTHER KAN-GNN VARIANTS

With the advent of the KAN architecture, some pioneering works (Kiamari et al., 2024; Zhang & Zhang, 2024; Bresson et al., 2024; De Carlo et al., 2024) have combined KAN with GNNs, leading to the development of various KAN-GNN variants. These works adopt a similar approach by using KAN as a feature transformation function within GNNs. The specific implementations and complete experimental results of these works have generally not been made public by the authors. We gather the existing experimental results to compare them with our proposed KAA, as shown in Table 7. The experimental results indicate that our proposed KAA-GAT achieves the optimal performance on these two node classification datasets Cora and CiteSeer. Additionally, KAA-GAT is the only model among these KAN-GNN variants that consistently outperforms classic GNN models.

Table 7: Results of accuracy (%) comparison between existing KAN-GNN variants and our proposed KAA-GAT.

| Dataset / Model | Cora | CiteSeer |
|---|---|---|
| GCN | 81.99 ±0.70 | 71.36 ±0.57 |
| GraphSAGE | 81.48 ±0.41 | 69.70 ±0.63 |
| GIN | 79.85 ±0.70 | 69.60 ±0.89 |
| KAGIN | 76.20 ±0.77 | 68.37 ±1.17 |
| KAGCN | 78.26 ±1.77 | 64.09 ±1.85 |
| GKAN | 81.20 | 69.40 |
| KAA-GAT | **83.80** ±0.49 | **73.10** ±0.61 |

### B.4 TIME AND SPACE ANALYSES

| | Cora | | CiteSeer | | PubMed | |
|---|---|---|---|---|---|---|
| | n (k) | t (ms) | n (k) | t (ms) | n (k) | t (ms) |
| GAT | 92.3 | 677.1 | 237.5 | 684.9 | 32.3 | 683.6 |
| KAA-GAT | 92.8 | 682.0 | 238.0 | 690.3 | 32.8 | 692.0 |

Table 8: Statistics of time and space cost.

Unlike KAN-based methods in other domains, KAA introduces a relatively small-scale KAN only in the scoring function part, which makes its time and space costs more manageable. In most cases in our experiments, the KAN we used consists of a single layer, with the spline order of the B-spline function set to 1 and a grid size of 1. This simple structure is sufficient to achieve satisfactory downstream results. Specifically, we collect statistics on the number $n$ of parameters and the time $t$ required for each training epoch for both GAT and KAA-GAT on the Cora, CiteSeer, and PubMed

datasets. The results in Table 8 show that most of the parameters in GNNs are used for feature transformation, allowing KAA to add less than 2% additional parameters. Moreover, the runtime increase is also under 2%. This indicates that in scenarios where KAA is applied, both the time and space costs are highly acceptable.

## B.5 KAA ON ADVANCED ATTENTIVE GNNS

To verify whether KAA remains effective on SOTA attentive GNNs, we select two advanced GAT-based models (SuperGAT (Kim & Oh, 2021) and HAT (Zhang et al., 2021)) and two advanced Transformer-based models (NAGformer (Chen et al., 2022) and SGFormer (Wu et al., 2024b)). These models integrate multiple techniques and feature relatively complex computation pipelines. We evaluate the effectiveness of KAA on these advanced attentive GNNs and present a comparison of the results between the original models and the KAA-enhanced models on node classification tasks. From Table 9, we can observe that even for various high-performance attentive GNN models, KAA still achieves significant performance improvements. KAA consistently delivers performance gains across all scenarios and achieves remarkable results on certain datasets. For instance, KAA enhances HAT's performance on the ogbn-arxiv dataset by approximately 3%, which is quite impressive. In many cases, the performance gains brought by KAA surpass the performance differences between different models, highlighting the value of incorporating KAA.

| | Cora | CiteSeer | PubMed | ogbn-arxiv | Computers | Photo |
|---|---|---|---|---|---|---|
| SuperGAT | 84.27 ±0.53 | 72.71 ±0.65 | 81.63 ±0.55 | 71.55 ±0.43 | 94.05 ±0.49 | 91.77 ±0.32 |
| KAA-SuperGAT | 84.45 ±0.23 | 73.08 ±0.28 | 81.72 ±0.43 | 74.01 ±0.67 | 94.65 ±0.78 | 92.20 ±0.22 |
| HAT | 83.67 ±0.39 | 72.31 ±0.23 | 79.67 ±0.45 | 70.87 ±0.65 | 93.77 ±0.46 | 90.55 ±0.34 |
| KAA-HAT | 84.01 ±0.40 | 72.55 ±0.64 | 79.90 ±0.42 | 73.55 ±0.76 | 93.80 ±0.37 | 91.65 ±0.40 |
| NAGphormer | 74.22 ±0.89 | 63.68 ±0.85 | 76.34 ±0.76 | ╲ | 91.23 ±0.78 | 90.04 ±0.22 |
| KAA-NAGphormer | 77.01 ±0.91 | 64.04 ±0.67 | 78.30 ±0.55 | ╲ | 91.42 ±0.81 | 91.00 ±0.45 |
| SGFormer | 75.46 ±0.78 | 67.79 ±0.26 | 79.89 ±0.64 | ╲ | 92.97 ±0.44 | 91.06 ±0.54 |
| KAA-SGFormer | 77.45 ±0.80 | 68.08 ±0.77 | 80.03 ±0.97 | ╲ | 93.08 ±0.79 | 91.33 ±0.59 |

Table 9: Accuracy (%) of KAA cooperated with advanced attentive GNNs on node classification.

