# OpenReview forum: "KAA: Kolmogorov-Arnold Attention for Enhancing Attentive Graph Neural Networks"
_ICLR.cc/2025/Conference — ICLR 2025 Poster_

### Official Review · Reviewer_nAy3 · 2024-10-16

**Soundness:** 3
**Presentation:** 3
**Contribution:** 2
**Rating:** 6
**Confidence:** 4

**Summary:**

In this paper, the authors unify scoring functions of current attentive GNNs and propose Kolmogorov-Arnold Attention (KAA), which integrates the Kolmogorov-Arnold Network (KAN) architecture into the scoring process. KAA enhances the performance of scoring functions across the board and can be applied to nearly all existing attentive GNNs. They also introduce Maximum Ranking Distance (MRD) to quantitatively estimate upper bounds in ranking errors for node importance. Experiments are provided on both node-level and graph-level tasks using various backbone models including KAA-enhanced scoring functions.

**Strengths:**

The Architecture of the proposed model is overall clearly written, and the background on KAN is well introduced.

Table 1 on the formulas are informative with many experiment benchmarks evaluated against.

**Weaknesses:**

Can the authors more clearly indicate why GAT’s attention mechanism which uses a learnable alpha weight via local neighborhoods is insufficient in terms of expressive power?

How does KAA attention compare to the more informative hierarchical attention mechanisms? The author should discuss this in the Related Work section. Here are recent relevant papers on this topic:

[WWW 2019] Heterogeneous Graph Attention Network. In The World Wide Web Conference (WWW '19). Association for Computing Machinery, New York, NY, USA, 2022–2032. https://doi.org/10.1145/3308558.3313562

[IEEE ICDM 2021] "Bi-Level Attention Graph Neural Networks," in 2021 IEEE International Conference on Data Mining (ICDM), Auckland, New Zealand, 2021 pp. 1126-1131.
doi: 10.1109/ICDM51629.2021.00133

**Questions:**

I have a concern that the novelty of the work is limited as it seems to be mainly using KAN (already proposed), with other already proposed components such as multi-head attention, and MLP like components. Can the authors more clearly distinguish their contributions from prior works?

Table 1 on the formulas are informative with many experiment benchmarks evaluated against. However, the GNN models seem to be quite outdated, and even the other benchmarks are mostly variant models of the author’s proposed KAA. Other KAN based benchmarks should also be provided.

---

> ### Author Response · Authors · 2024-11-20
> **Response to Reviewer nAy3 (1/2)**
>
> Dear reviewer nAy3,
> We hope our point-to-point responses can address your concerns and provide you with better clarification of the contributions and value of our work.
>
> 1. (Weakness 1) Clarification for insufficiency of existing attention mechanism.
>
> The goal of GAT is to "adaptively" learn the alpha weights of neighboring nodes for a given central node. Our research examines whether GAT's method of learning these weights is sufficiently "adaptive." Why do we consider GAT's scoring function to lack expressiveness and be insufficiently "adaptive"? This is because GAT essentially performs a linear transformation of the features of neighboring nodes to compute their importance scores. Such transformations are not "rich" enough and inherently impose many implicit constraints. A toy example illustrates this limitation: suppose a node has three neighbors, $x_1$, $x_2$, and $x_3$, with feature values of 1, 2, and 3, respectively. Using GAT's linear scoring function $s = ax + b$, regardless of how the learnable parameters $a$ and $b$ are chosen, either $x_1$ or $x_3$ will always receive the highest score, while $x_2$ can never achieve the highest attention score. This demonstrates the lack of expressiveness in the scoring function, as it fails to adaptively assign importance based on the unique characteristics of the features. Of course, real-world problems are far more complex than this toy example. In our paper, we provide a detailed analysis and discussion of these issues. KAA is designed to universally enhance the expressiveness and adaptability of the scoring function, enabling various attentive GNNs to truly learn the alpha weights they aim to capture.
>
> 2. (Weakness 2) Comparison with hierarchical attention mechanism.
>
> Most existing methods focus on determining what constitutes important attention information, with advanced techniques excelling at capturing this crucial information through the attention mechanism. The core of these methods lies in guiding the model to focus on what type of information should be captured by the attention mechanism to make it more informative. The hierarchical attention mechanism you mentioned also falls within this category of approaches. KAA, on the other hand, emphasizes the actual learning process and improves the construction of the attention mechanism itself. Our goal is not to prescribe what type of attention should be captured but to use theoretical analysis and experimental results to demonstrate how to design a scoring function when aiming to model a specific attention distribution. This ensures that your model can truly and adaptively capture the desired attention distribution effectively. This means that, regardless of the type of attention distribution being learned, KAA can enhance the success rate of modeling that distribution. As you mentioned, the hierarchical attention mechanism is intuitive and meaningful in its design. KAA can further enhance the practical expressiveness of such methods. We conducted a simple validation on the two methods you referred to, and the results are presented in the table below.
>
>
>
> |           | IMDB(Micro-F1%) | MUTAG (Accuracy %) |
> | --------- | --------------- | ------------------ |
> | HAN       | 55.73           | \                  |
> | KAA-HAN   | 56.39           | \                  |
> | BAGNN     | \               | 87.81              |
> | KAA-BAGNN | \               | 88.10              |
>
> Table. Comparison results of KAA on hierarchical attention methods.
>
>
>
> The experimental results show that KAA consistently provides performance gains, highlighting its ability to enable attentive GNNs to more effectively learn the attention weights they aim to capture. In addition, we will include a similar discussion in the paper to clarify the relationship between KAA and other methods.

---

> ### Author Response · Authors · 2024-11-20
> **Response to Reviewer nAy3 (2/2)**
>
> 3. (Question 1) Contributions of our work.
>
> Our paper makes significant contributions to the design and development of attentive GNN models. First, we unify the existing paradigms of attentive GNNs, covering nearly all GAT-based and Transformer-based models. Building on this foundation, we innovatively identify and explore the issue of insufficient ranking expressiveness in attentive GNNs and develop a comprehensive theoretical framework to address this problem. Using our framework, we can not only quantitatively assess the most critical scoring capabilities of attentive GNNs but also intuitively compare the expressiveness across different methods. Ultimately, we find that incorporating a small-scale KAN network effectively resolves this issue and propose the KAA method. The simplicity and efficiency of KAA make it an ideal plug-in, capable of universally enhancing all attentive GNNs to better capture the attention distributions they aim to learn. Our contributions to the KAN field are also notable. We introduce KAN into the scoring function and achieve significant success, offering valuable insights for other researchers on how to apply KAN effectively. More importantly, we innovate by comparing the theoretical expressive capabilities of MLP and KAN. This novel perspective will greatly inspire the KAN community and help researchers determine when to use KAN versus MLP. Therefore, our work has made substantial theoretical contributions to both the attentive GNN and KAN communities. It would be unfair to dismiss the innovation of our paper due to the fundamental nature of our solution.
>
> 4. (Question 2) Advanced baselines.
>
> We have presented the performance results of KAA on advanced attentive GNNs in the Global Response. The experimental results demonstrate that KAA consistently provides significant performance gains, even when applied to advanced models. Additionally, a comparison with other KAN-based baselines has already been provided in Appendix B.3. This comparison includes all existing KAN variants for which we are able to collect either code or results applicable to graph data.
>
> We hope our explanation can dispel your concerns and motivate you to reconsider the value of our paper. If you have any other questions or concerns, please feel free to let us know.

---

> ### Author Response · Authors · 2024-11-26
> **Looking Forward to Your Feedback**
>
> Dear reviewer nAy3,
> As there is only one day remaining to submit the revised paper, we would greatly appreciate it if you could kindly take some time to respond to our rebuttal. We would also like to confirm whether our explanations have adequately addressed your questions and concerns. Thank you for your time and consideration.

---

> > ### Comment · Reviewer_nAy3 · 2024-11-27
> >
> > Thanks for your thorough response to my questions. Upon further consideration, I feel this work does have some merit in improvement state of the art, though the novelty is incremental and already an exhaustively studied area. I will raise my score by 1 point to a 5.

---

> > > ### Author Response · Authors · 2024-11-27
> > > **Further Response to Reviewer nAy3**
> > >
> > > Dear reviewer nAy3,
> > > Thank you for your feedback. We would also like to present some additional remarks on the contributions of our paper and the field of attentive GNNs.
> > >
> > > 1. Main contributions of our work.
> > >
> > > The theoretical contributions of our paper are significant. In the field of attentive GNNs, we have pioneered a unified perspective on the ranking problem within existing scoring functions. **This issue is so fundamental that prior research on the implicit constraints of attentive GNNs can be viewed as special cases of the ranking problem.** Moreover, our analytical framework broadens the theoretical applicability to all types of attentive GNNs, including both GAT-based and Transformer-based models. This framework has the potential to inspire the design of many new attentive GNNs. Additionally, **our work addresses an important problem in the KAN community: how to compare the theoretical expressive power of MLP and KAN.** From the perspective of parameter limitations, we have quantitatively characterized the differences between KAN and MLP. Our theoretical approach provides a strong foundation for the KAN community and offers guidance on the most appropriate scenarios for applying KAN. Overall, our work provides a solid theoretical foundation for both the attentive GNN field and the KAN community, and is expected to inspire a series of subsequent studies.
> > >
> > > 2. About attentive GNNs.
> > >
> > > The field of attentive GNNs discussed in this paper, while widely studied, has encountered new opportunities and challenges in recent years. One promising direction is the development of graph foundation models (GFM) [1,2], which can adapt to a variety of tasks, much like large language models. Many existing GFM models [3,4,5] adopt attentive GNNs. Drawing from the success of large language models, the attention mechanism has proven to be crucial for modeling cognition [6]. Therefore, it is reasonable to expect that as GNNs evolve towards greater generalization, attentive GNNs will continue to play a pivotal role. To build a high-quality GFM similar to large language models, the expressive power of the underlying model is of unprecedented importance. The KAA we propose universally enhances the expressive power of attentive GNNs. **This fundamental and versatile technique can serve as a cornerstone for the next generation of attentive GNNs, playing a key role in both the construction of GFMs and the continued development of attentive GNNs.**
> > >
> > > Thank you again for your feedback. We also hope that our further responses would motivate you to reconsider the value of our paper.
> > >
> > > **Reference**
> > >
> > > [1] Towards Graph Foundation Models: A Survey and Beyond.
> > >
> > > [2] A Survey on Self-Supervised Pre-Training of Graph Foundation Models: A Knowledge-Based Perspective.
> > >
> > > [3] Graphany: A Foundation Model for Node Classification on Any Graph.
> > >
> > > [4] Building Transportation Foundation Model via Generative Graph Transformer.
> > >
> > > [5] GOFA: A Generative One-For-All Model for Joint Graph Language Modeling.
> > >
> > > [6] Attention Mechanism in Neural Networks: Where It Comes and Where It Goes.

---

> > > > ### Comment · Reviewer_nAy3 · 2024-11-29
> > > >
> > > > Thanks for the details and they have addressed my concerns. I have raised my score to 6.

---

> > > > > ### Author Response · Authors · 2024-11-29
> > > > >
> > > > > Thank you for your support of our work. Your valuable suggestions have made our work better.

---

### Official Review · Reviewer_nWMV · 2024-11-02

**Soundness:** 3
**Presentation:** 3
**Contribution:** 2
**Rating:** 8
**Confidence:** 4

**Summary:**

leveraging Kolmogorov-Arnold Network (KAN), the authors proposed to use the KA attention to enhance attention-based GNN architectures. The paper provides extensive experiments to show that attention-based GANs are enhanced on both node-level and graph-level tasks across a wide range of benchmarks.

**Strengths:**

1. The authors provides a thorough analysis and recap over KAN and its application and components on attention-based GNNs, for example the expressive power of scoring functions and theoretical bounds of transform-based attention, MLP-based attention, and the proposed KAA. The theoretical analysis shows that KAA exhibits the strongest expressive power.
2. Extensive experiments across a wide range of benchmarks for node-level and graph-level tasks show that KAA could provide moderate and consistent improvements.

**Weaknesses:**

1. The author provides a very interesting plugins for attention-based GNNs and shows relatively good improvement. I personally like this idea and it shows potential of KAA on graph applications. however, one major concern is that nowadays, attention is still considered not necessary for GNN architectures and many SOTA architectures did not utilize attention mechanism within their architecture design, e.g. [1] for node classification and [2] for link prediction. As a result, it would be better if there were more comparison with other SOTA GNN models who did not include attention mechanisms. This is a kind suggestion and new experimental results is not necessary.
2. There seems to be lack of empirical analysis/ablation studies for different ranking methods proposed in section 4.2.2.

[1] Li, Guohao, et al. "Training graph neural networks with 1000 layers." International conference on machine learning. PMLR, 2021.
[2] Zhang, Muhan, and Yixin Chen. "Link prediction based on graph neural networks." Advances in neural information processing systems 31 (2018).

**Questions:**

1. Could you provide an empirical ablation study over the ranking methods proposed in section 4.2.2 to demonstrate the effectiveness of proposed MRD?

---

> ### Author Response · Authors · 2024-11-20
> **Response to Reviewer nWMV (1/2)**
>
> Dear reviewer nWMV,
> We hope our point-to-point responses can address your concerns and provide you with better clarification of the contributions and value of our work.
>
> 1. (Weakness 1) Impact and comparison with advanced techniques.
>
> In recent years, GNNs have been continuously evolving toward greater generalization and robustness. One popular direction is the development of graph foundation models (GFM) [1,2] that can adapt to various tasks, similar to large language models. Most existing GFM models [3,4,5] adopt attentive GNNs. From the success of large language models, the attention mechanism appears to be important for modeling cognition [6]. Therefore, we can believe that as GNNs evolve toward greater generalization, attentive GNNs will continue to play a crucial role. In addition, the attention mechanism often has an orthogonal relationship with other techniques, and combining these techniques can often lead to better downstream performance. As mentioned in papers [7,8], high-order cooperation is one such example, and many improved attentive GNNs [9,10] have also adopted similar ideas, achieving promising results. KAA strengthens the foundation of attentive GNNs and will play a significant role in paving the way for GNNs to achieve greater generalization and enhanced cognition.
>
> Your suggestion to incorporate advanced technologies is very reasonable. In fact, the KAA technique is not in conflict with other non-attention techniques, but these orthogonal methods are difficult to compare directly. Therefore, we select several of the most advanced and sophisticated GNN models and demonstrate that KAA consistently provides stable and satisfactory improvements when applied to these SOTA techniques. We have presented the performance results of KAA on advanced attentive GNNs in the Global Response. The experimental results show that KAA consistently provides significant performance gains even on advanced models.

---

> ### Author Response · Authors · 2024-11-20
> **Response to Reviewer nWMV (2/2)**
>
> 2. (Weakness 2 \& Question 1) Empirical guarantee of MRD.
>
> Thanks for your valuable suggestions, and the experimental validation of MRD's effectiveness is highly meaningful. To achieve this target, we design a synthetic node regression task to validate this. Specifically, we first construct an Erdős-Rényi (ER) random graph $G(N, p)$ with $N$ nodes and an edge probability $p$. Next, we randomly generate $d$-dimensional Gaussian features $x_i \in \mathcal{N}(0, 1)^d$ for each node. For a node $v_i$ with $n_i$ neighbors, we randomly assign a target permutation $\pi_i: \{1,...,n_i\} \to \{1,...,n_i\}$ to its neighbors, serving as the target ranking for $v_i$ as well as the target scores of its neighbors relative to $v_i$. After normalizing the target scores, we obtain the target aggregation coefficients $\alpha_{i,j}$ for the neighbor node $v_j$ of the central node $v_i$. Finally, we assume the regression value for each node to be $h_i = \sum \alpha_{i,j} x_j$. We use GAT (representing Linear Transformation-Based Attention), GATv2 (representing MLP-Based Attention), and KAA-GAT (representing Kolmogorov-Arnold Attention) to perform single-layer node value regression prediction. The structural difference between them lies only in the scoring functions: Linear Transformation, MLP, and KAN, respectively. Their MRD satisfies the inequality $MRD_{KAA-GAT} < MRD_{GATv2} < MRD_{GAT}$. On a graph with $d = 10$, $N = 100$, and $p = 0.1$, the relative values of the actual RD and error $E$ for each method are shown in the table below.
>
>
>
> |        | GAT   | GATv2 | KAA-GAT |
> | ------ | ----- | ----- | ------- |
> | E (%)  | 100.0 | 92.7  | 62.3    |
> | RD (%) | 100.0 | 85.6  | 54.3    |
>
> Table. Relative error on $G(100, 0.1)$.
>
>
>
> On a graph with $d = 10$, $N = 1000$, and $p = 0.01$, the relative values of the actual RD and error $E$ for each method are shown in the table below.
>
>
>
> |        | GAT   | GATv2 | KAA-GAT |
> | ------ | ----- | ----- | ------- |
> | E (%)  | 100.0 | 84.2  | 45.8    |
> | RD (%) | 100.0 | 80.3  | 39.6    |
>
> Table. Relative error on $G(1000, 0.01)$.
>
>
>
> From the experimental table, we can observe that KAA-GAT achieves the smallest MRD, with the smallest actual RD and error as well. GATv2 has the second-lowest MRD, and its results also rank second in terms of RD and error. Next, we base our modifications on KAA-GAT, reducing the hidden layer dimensions of KAN to half and a quarter of the original size to decrease its MRD. The results for these three models on two graphs are shown below:
>
>
>
> |        | KAA-GAT | KAA-GAT (1/2) | KAA-GAT (1/4) |
> | ------ | ------- | ------------- | ------------- |
> | E (%)  | 100.0   | 117.2         | 126.3         |
> | RD (%) | 100.0   | 125.0         | 132.4         |
>
> Table. Relative error on $G(100, 0.1)$.
>
>
>
> |        | KAA-GAT | KAA-GAT (1/2) | KAA-GAT (1/4) |
> | ------ | ------- | ------------- | ------------- |
> | E (%)  | 100.0   | 127.9         | 133.2         |
> | RD (%) | 100.0   | 131.3         | 139.1         |
>
> Table. Relative error on $G(1000, 0.01)$.
>
>
>
> At this point, it can be observed that all models with smaller MRD achieve better performance, which demonstrates the effectiveness of the MRD metric. This supports the idea that minimizing MRD directly correlates with improved model performance. We will provide a more detailed description of this synthetic experiment, including a comprehensive set of results, in the appendix later.
>
> We hope our explanation can dispel your concerns. If you have any other questions or concerns, please feel free to let us know.
>
> **Reference**
>
> [1] Towards Graph Foundation Models: A Survey and Beyond.
>
> [2] A Survey on Self-Supervised Pre-Training of Graph Foundation Models: A Knowledge-Based Perspective.
>
> [3] Graphany: A Foundation Model for Node Classification on Any Graph.
>
> [4] Building Transportation Foundation Model via Generative Graph Transformer.
>
> [5] GOFA: A Generative One-For-All Model for Joint Graph Language Modeling.
>
> [6] Attention Mechanism in Neural Networks: Where It Comes and Where It Goes.
>
> [7] Training Graph Neural Networks with 1000 Layers.
>
> [8] Link Prediction Based on Graph Neural Networks.
>
> [9] Multi-Hop Attention Graph Neural Networks.
>
> [10] Inductive Link Prediction with Interactive Structure Learning on Attributed Graph.

---

> > ### Comment · Reviewer_nWMV · 2024-11-25
> >
> > Thanks for the authors' response and my concerns are well-addressed. I am willing to raise my score to 8.

---

> > > ### Author Response · Authors · 2024-11-26
> > >
> > > Thank you for your support of our work. Your valuable suggestions have made our work better.

---

### Official Review · Reviewer_kkV1 · 2024-11-03

**Soundness:** 3
**Presentation:** 3
**Contribution:** 2
**Rating:** 6
**Confidence:** 4

**Summary:**

This paper introduces a novel scoring function, Kolmogorov-Arnold Attention (KAA), which integrates the Kolmogorov-Arnold Network (KAN) architecture to enhance the performance of scoring functions in attentive Graph Neural Networks (GNNs). The authors propose a unified framework for scoring functions and introduce the Maximum Ranking Distance (MRD) to quantitatively measure their expressive power. Extensive experiments demonstrate that KAA-enhanced models outperform their original counterparts across various tasks and backbone architectures, with significant improvements in some cases.

**Strengths:**

### Strengths:

1. The paper presents a creative approach to enhancing GNNs by integrating the Kolmogorov-Arnold Network into the scoring process, which is a novel contribution to the field of graph neural networks.

2. The authors not only propose a new model but also provide a rigorous theoretical analysis of its expressive power compared to existing methods, backed by extensive experimental results that validate the effectiveness of KAA.

3. The paper demonstrates substantial performance improvements over some baselines on both node-level, link-level, and graph-level tasks, highlighting the practical applicability and benefits of the proposed KAA framework.

**Weaknesses:**

### Weaknesses:

1. I have one concern that despite the interesting introduction of KAA into the scoring function in graph attention networks, it is not that big to modify the architecture of attention computing mechanism for existing GNN community since there have been various GNN architectures beyond attention functions in the past years.

2. Then in the experiments, the authors only compare with some **old and weak** baselines with shallow layers ( $\le$ 4 layers), which further support my claim in the first concern.

**Questions:**

Check the questions below.

---

> ### Author Response · Authors · 2024-11-20
> **Response to Reviewer kkV1**
>
> Dear reviewer kkV1,
> We hope our point-to-point responses can address your concerns and provide you with better clarification of the contributions and value of our work.
>
> (Weakness 1) Potential impact for GNN community.
>
> Your interpretation of the development of the entire GNN community is very reasonable. However, overall, attentive GNN models are still widely applied in various scenarios. In recent years, GNNs have been continuously evolving toward greater generalization and robustness. One popular direction is the development of graph foundation models (GFM) [1,2] that can adapt to various tasks, similar to large language models. Most existing GFM models [3,4,5] adopt attentive GNNs. From the success of large language models, the attention mechanism appears to be important for modeling cognition [6]. Therefore, we can believe that as GNNs evolve toward greater generalization, attentive GNNs will continue to play a crucial role. Our work strengthens the foundation of attentive GNNs, and we hope that KAA can serve as an optional learner in the future, easily integrated into various existing attentive GNN models.
>
> (Weakness 2) Advanced baselines.
>
> We have presented the performance results of KAA on advanced attentive GNNs in the Global Response. Experimental results show that KAA consistently provides significant performance gains even on advanced models. Moreover, for Graph Transformers, the number of layers often exceeds 4 in many scenarios, especially in graph-level tasks. This indicates that the results and conclusions of the paper are not limited to shallow models.
>
> We hope our explanation can dispel your concerns. If you have any other questions or concerns, please feel free to let us know.
>
> **Reference**
>
> [1] Towards Graph Foundation Models: A Survey and Beyond.
>
> [2] A Survey on Self-Supervised Pre-Training of Graph Foundation Models: A Knowledge-Based Perspective.
>
> [3] Graphany: A Foundation Model for Node Classification on Any Graph.
>
> [4] Building Transportation Foundation Model via Generative Graph Transformer.
>
> [5] GOFA: A Generative One-For-All Model for Joint Graph Language Modeling.
>
> [6] Attention Mechanism in Neural Networks: Where It Comes and Where It Goes.

---

> ### Comment · Reviewer_kkV1 · 2024-11-26
>
> Dear authors,
>
> Thanks for your responses. For weakness 1, I understood your claim, and admit your contribution. However, I would like to keep my point due to that the attention mechanism relating to KAA is not that convincing in terms of performance of graph problems
>  considering so many GNN architectures have been proposed.
>
> Best,
>
> Reviewer

---

> > ### Author Response · Authors · 2024-11-29
> >
> > Thank you for your support of our work. Your valuable suggestions have made our work better.

---

### Official Review · Reviewer_w7oB · 2024-11-04

**Soundness:** 3
**Presentation:** 3
**Contribution:** 2
**Rating:** 6
**Confidence:** 4

**Summary:**

This paper introduces Kolmogorov-Arnold Attention (KAA), with a novel scoring method for graph attention. KAA enhances the performance of scoring functions and can be applied to nearly all existing attentive GNNs. To further analyze the expressive power of scoring functions in various graph attention mechanism, this paper introduces Maximum Ranking Distance (MRD) to quantitatively estimate their upper bounds in ranking errors. Extensive experiments on both node-level and graph-level tasks show the KAA can outperforms their original counterparts.

**Strengths:**

1. Exploring KAN for graph attention is novel.
2. Based on the analysis tool of graph attention in GATv2 [1], this paper proposes more powerful tools, Importance Ranking and Ranking Distance, to analyze the expressive power of graph attention.
3. The KAA is extensively evaluated in various downstream tasks.

**Weaknesses:**

1. The motivation to redesign the graph attention with KAN is not convincing. Current graph attention and graph Transformer have provided very good performance. The authors are encouraged to illustrate when the proposed KAN will significantly outperform current graph attention, including GATv2 [1], and graph Transformer.
2. I am confused by MRD theory. If there exists an optimum ranking for each node, and KAA always provides a good ranking that is very close to the optimum ranking, does KAA still needs multi-head attention? It seems that different attention heads will provide very similar attention. However, multi-head attention has been proven to be a very successful tech in attention.
3. The assumption that "the selected central node is connected to all other nodes in the graph" is not realistic. Even graph Transformers [3] are trying to incorporate various graph structures, which violate the assumption.
4. As claimed by the authors, the expressive power of KAA is much better than GAT and GATv2. However, better expressive power could result in severe overfitting [1]. I wonder how the authors tackle the severe overfitting issue in KAA.
5. The inclusion of KAN would significantly increase the computational cost. Thus the authors are encouraged to compare its computational cost (, including both time and space) with current graph attention and graph Transformer.
6. The comparisons with baselines could be unfair. The authors do not choose the recommended hyper-parameters as in the original paper or tune better results. For example, the dropout ratio in the GAT [2] paper is 0.6, and [2] achieves better results than the scores reported in this paper, under the same data split settings.

I will raise my score if my concerns are properly addressed.

---

[1] How Attentive are Graph Attention Networks? In ICLR 2022.

[2] Graph Attention Networks. In ICLR 2018.

[3] Do Transformers Really Perform Bad for Graph Representation? In NeurIPS 2021

**Questions:**

Please refer to Weaknesses.

---

> ### Author Response · Authors · 2024-11-20
> **Response to Reviewer w7oB (1/2)**
>
> Dear reviewer w7oB,
> We hope our point-to-point responses can address your concerns and provide you with better clarification of the contributions and value of our work.
>
> 1. (Weakness 1) Motivations to design attentive GNNs with KAN.
>
> We have presented the performance results of KAA on advanced attentive GNNs in the Global Response. A comparison between KAA and GATv2 has already been provided in Table 6 of the paper's appendix. In all cases, KAA demonstrates superior performance. In fact, KAA is not at odds with advanced attentive GNN techniques and does not lose effectiveness due to their strong performance. This is because existing methods focus more on determining what constitutes important attention information, and advanced methods often excel at capturing such crucial information through the attention mechanism. KAA, on the other hand, enhances the actual construction process of the attention mechanism. This means that regardless of the type of attention distribution being learned, KAA can improve the success rate of accurately modeling that distribution. Therefore, despite the significant advancements in existing attentive GNNs, KAA can still universally provide performance improvements, as it represents an enhancement from a different perspective.
>
> 2. (Weakness 2) Multi-head attention with MRD theory.
>
> The impact of multi-head attention on the theoretical expressive power of a model depends on the specific model architecture [1]. However, in most practical cases, it generally has a positive effect on downstream tasks. A toy example illustrating this is when we disregard the normalization of attention scores in GAT. If there are two attention heads with coefficients $W_1$ and $W_2$, producing results $H_1$ and $H_2$, respectively, and the final result is $H = H_1 + H_2$, we can equivalently find a single-head attention coefficient $W_3 = W_1 + W_2$ that directly yields $H$. This suggests that, in the context of attentive GNNs, multi-head attention is often used more as a practical trick. It helps reduce task complexity and alleviates the optimization process [2]. Your understanding of KAA and our proposed MRD framework is absolutely correct. While multi-head attention offers little enhancement in theoretical expressive power, it remains a valuable practical trick to improve downstream results [2]. This is why we still employ multi-head attention in KAA.
>
> 3. (Weakness 3) Question about the assumption.
>
> The reason we assume that the central node is connected to all other nodes is to ensure simplicity in expression and to minimize the introduction of new symbols. We apologize if this assumption seemed overly strong. In reality, our theoretical analysis only imposes certain requirements on the size of the alignment matrix $P$. As referenced in Formula 27 in the Appendix, the result holds as long as the number of rows in $P$ exceeds the number of columns. This implies that the error bounds in our analysis hold as long as the central node has more neighbors at a given layer than its hidden dimension, which is easily satisfied in real-world scenarios. For instance, the Bio and Chem datasets inherently have only 2-dimensional features, and on the Cora dataset, GAT uses a hidden dimension of 8. Therefore, the theoretical analysis presented in the paper is much more broadly applicable than the conditions assumed for simplicity. The seemingly strong assumptions were made solely to facilitate easier reading and understanding. Thank you for your suggestion, and we have included additional clarifications in our latest revision.
>
> 4. (Weakness 4) Solution to overfitting.
>
> Your concern about more expressive models being more prone to overfitting is very reasonable. KAA not only has a strong theoretical upper bound but also leverages simple techniques in downstream tasks to mitigate overfitting, thereby achieving robust performance. In our experiments, we found that simply increasing the dropout rate and appropriately reducing the hidden layer dimension can yield excellent results with KAA, without requiring additional regularization. This indicates that KAA is not only a method with a high theoretical ceiling but also one that is simple to implement and easy to tune, making it highly adaptable to a wide range of real-world scenarios.

---

> ### Author Response · Authors · 2024-11-20
> **Response to Reviewer w7oB (2/2)**
>
> 5. (Weakness 5) Time and space cost of KAA.
>
> Unlike KAN-based methods in other domains, KAA introduces a relatively small-scale KAN only in the scoring function part, which makes its time and space costs more manageable. In most cases in our experiments, the KAN we used consists of a single layer, with the spline order of the B-spline function set to 1 and a grid size of 1. This simple structure is sufficient to achieve satisfactory downstream results. Specifically, we collect statistics on the number $n$ of parameters and the time $t$ required for each training epoch for both GAT and KAA-GAT on the Cora, CiteSeer, and PubMed datasets. The results are shown in the table below.
>
>
>
> |         | Cora  |        | CiteSeer |        | PubMed |        |
> | ------- | ----- | ------ | -------- | ------ | ------ | ------ |
> |         | n (k) | t (ms) | n (k)    | t (ms) | n (k)  | t (ms) |
> | GAT     | 92.3  | 677.1  | 237.5    | 684.9  | 32.3   | 683.6  |
> | KAA-GAT | 92.8  | 682.0  | 238.0    | 690.3  | 32.8   | 692.0  |
>
> Table. Statistics of time and space cost.
>
>
>
> The results in the table show that most of the parameters in GNNs are used for feature transformation, allowing KAA to add less than 2% additional parameters. Moreover, the runtime increase is also under 2%. This indicates that in scenarios where KAA is applied, both the time and space costs are highly acceptable.
>
> 6. (Weakness 6) Baseline setting.
>
> In Table 1, the performance of GAT on some datasets is slightly lower than the results reported in the original paper. This discrepancy arises because we standardized the selection of hidden layer dimensions for all models from the range [32, 64, 128, 256] (as mentioned in Line 427 of the original paper). Most existing models use hidden layer dimensions within this range. However, GAT's optimal hidden layer dimension on Cora, CiteSeer, and PubMed is 8. To address this issue, we expanded the hidden layer dimension selection to include [8, 16] and re-tuned the parameters for all models. We found that only GAT and KAA-GAT achieved better results on some node classification datasets with these additional dimensions. The updated results are provided in the table below.
>
>
>
> |         | Cora         | CiteSeer     | PubMed       | Computers    |
> | ------- | ------------ | ------------ | ------------ | ------------ |
> | GAT     | 83.07 (0.45) | 72.61 (0.53) | 79.04 (0.89) | 92.89 (0.40) |
> | KAA-GAT | 83.87 (0.66) | 73.53 (0.31) | 79.45 (0.72) | 93.55 (0.27) |
>
> Table. Updated experimental results.
>
>
>
> This adjustment leads to comparable performance improvements for both GAT and KAA-GAT, reinforcing the consistency of our previous experimental conclusions that KAA continues to deliver performance gains across all cases. Thank you for your detailed feedback. We have thoroughly rechecked the baseline results from our experiments against those reported in the original paper and can confirm that our results consistently outperform the originals. (Note that the specialized tricks of each baseline model have been removed, so the comparison is based solely on the results that include only the core model.)
>
> We hope our explanation can dispel your concerns. If you have any other questions or concerns, please feel free to let us know.
>
> **Reference**
>
> [1] Repulsive Attention: Rethinking Multi-head Attention as Bayesian Inference.
>
> [2] On the Optimization and Generalization of Multi-head Attention.

---

> > ### Comment · Reviewer_w7oB · 2024-11-26
> >
> > I would like to thank the authors for their rebuttal. While I appreciate the effort, some of my concerns remain unresolved:
> > 1. (W1) Simply demonstrating better empirical performance is not sufficient. Especially, when you are trying to improve upon a well-studied and widely-used method.
> > 2. (W2) The outputs from different attention heads are concatenated in GAT, and thus you cannot write $W_3 = W_2 + W_1$. If you check the attention scores from GAT, you can find that the attention distributions from different attention heads are different. This aligns with the observations in [1], "Multi-head attention allows the model to jointly attend to information from different representation subspaces at different positions". However, your theoretical results seem to contradict this established understanding, raising concerns about the applicability of your analysis to actual model design.
> > 3. (W3) Does the theoretical results hold, or partially hold in other graphs? I understand that some assumptions are required to simplify the problem to get interesting theoretical results. However, the fully connected graph loses all graph structures, and thus it is not a good assumption.
> >
> > [1] Attention Is All You Need.

---

> > > ### Author Response · Authors · 2024-11-26
> > > **Further Response to Reviewer w7oB (2/2)**
> > >
> > > 3. (Weakness 3) Question about the assumption.
> > >
> > > **First of all, our theoretical analysis does not require the involved graphs to be fully connected and is applicable to almost all types of graphs.** In fact, our theoretical conclusion is quite intuitive: central nodes with more neighbors are more likely to encounter ranking problems in the scoring function. In the original paper, we chose a central node with the most neighbors (i.e., a node connected to all other nodes) for analysis. However, this does not imply that the original graph must be fully connected. In our rebuttal response, we have already clarified the scope of applicability of our theory. The occurrence of the ranking problem is unrelated to the connectivity of the central node within the entire graph. **What matters is the relative size of the central node’s number of neighbors and its hidden layer dimension.** Specifically, as referenced in Formula 27 in the Appendix, our theoretical conclusions hold as long as the number of rows in  $P$ exceeds the number of columns. This implies that the error bounds in our analysis hold as long as the central node has more neighbors at a given layer than its hidden dimension—a condition easily satisfied in real-world scenarios. For example, the Bio and Chem datasets inherently have only 2-dimensional features, and on the Cora dataset, GAT uses a hidden dimension of 8. Therefore, our conclusion can be applied to almost all graphs. The only numerical modification in the conclusion is that the number of neighbor nodes is changed from $N$ to the actual number of neighbors $n_i$ of the central node, while all other elements remain unchanged.
> > >
> > > We hope our explanation can dispel your concerns and motivate you to reconsider the value of our paper. If you have any other questions or concerns, please feel free to let us know.
> > >
> > > **Reference**
> > >
> > > [1] On the Optimization and Generalization of Multi-head Attention

---

> > > > ### Comment · Reviewer_w7oB · 2024-11-28
> > > >
> > > > I would like to thank the authors for their rebuttal. I will increase my rating to 6.

---

> > > > > ### Author Response · Authors · 2024-11-28
> > > > >
> > > > > Thank you for your support of our work. Your valuable suggestions have made our work better.

---

> ### Author Response · Authors · 2024-11-26
> **Looking Forward to Your Feedback**
>
> Dear reviewer w7oB,
> As there is only one day remaining to submit the revised paper, we would greatly appreciate it if you could kindly take some time to respond to our rebuttal. We would also like to confirm whether our explanations have adequately addressed your questions and concerns. Thank you for your time and consideration.

---

> ### Author Response · Authors · 2024-11-26
> **Further Response to Reviewer w7oB (1/2)**
>
> Dear reviewer w7oB,
> We apologize for any misunderstanding caused by our unclear responses. Regarding your further concerns, we would like to clarify the following points.
>
> 1. (Weakness 1) Motivations to design attentive GNNs with KAN.
>
> Regarding the improvement of KAA over existing attentive GNNs, **we have demonstrated its effectiveness from three perspectives: intuition, theory, and experiments.** First, from an intuitive perspective, existing attentive GNN methods (both GAT-based and Transformer-based) focus primarily on identifying what constitutes important attention information. Advanced methods often excel at capturing such crucial information through the attention mechanism. These methods define what good attention looks like. KAA, on the other hand, takes a different approach. Once the model has identified the type of attention to learn, KAA allows the model to better capture this attention distribution. As a result, KAA can universally enhance the performance of existing methods. Second, from a theoretical perspective, regardless of how complex the logic and ideas behind existing methods are, their scoring functions will always take the unified form as shown in Equation 6 of the original paper. The real challenge in training then becomes how to enable this scoring function to learn the attention distribution we want. Section 4.2 of the paper rigorously proves, from a theoretical standpoint, why KAA leads to improvements across various forms of scoring functions. Thus, KAA is also theoretically effective for attentive GNNs. Finally, we have validated the effectiveness of KAA through experiments. Whether in the experiments presented in the main body of the paper or the additional experiments on advanced baselines provided in the rebuttal, KAA consistently improves the original model across all settings. In some cases, the improvement is even significant, reaching up to 20%. In summary, the effectiveness of KAA for attentive GNNs, whether classical or advanced, has been demonstrated comprehensively through intuition, theory, and experiments.
>
> 2. (Weakness 2) Multi-head attention with MRD theory.
>
> We apologize for the lack of clarity in our response and for the inappropriate example we provided regarding GAT. **We fully agree with your understanding of multi-head attention, as it enhances the model's expressive power. Our theoretical analysis is also consistent with the general understanding of multi-head attention.** The original question you raised concerns whether, given KAA's strong theoretical expressive power, it still requires multi-head attention to further enhance its expressiveness, and whether the attention distributions learned by different heads will be very similar. We will answer this question from two perspectives. First, while KAA has strong theoretical expressive power, fully realizing this capability requires certain conditions on the dimensionality of KAA’s hidden layers. However, **in practical training, to prevent overfitting or to accelerate convergence, the hidden layer dimensionality may not meet the theoretical requirements**. In this case, multi-head attention can be used to enhance KAA’s expressive power and improve the performance of downstream tasks. Second, like all other learning models, there is a gap between KAA’s theoretical expressive power and its actual performance. **In this context, multi-head attention helps to reduce task complexity and alleviate the optimization process [1]**. In our practical training, multi-head attention indeed provides a performance boost to KAA, and the attention distributions learned by different heads are distinct. That is why we apply the multi-head technique to KAA.

---

### Author Response · Authors · 2024-11-20
**Global Response**

Dear all reviewers,
We appreciate your valuable comments on our work. We provide the additional experimental results based on feedback. Furthermore, we have already uploaded the latest revision of our paper.

**KAA on advanced attentive GNNs**

To verify whether KAA remains effective on SOTA attentive GNNs, we select two advanced GAT-based models (SuperGAT [1] and HAT [2]) and two advanced Transformer-based models (NAGformer [3] and SGFormer [4]). These models integrate multiple techniques and feature relatively complex computation pipelines. We evaluate the effectiveness of KAA on these advanced attentive GNNs and present a comparison of the results between the original models and the KAA-enhanced models on node classification tasks in the table below.



|                | Cora         | CiteSeer     | PubMed       | ogbn-arxiv   | Computers    | Photo        |
| -------------- | ------------ | ------------ | ------------ | ------------ | ------------ | ------------ |
| SuperGAT       | 84.27 (0.53) | 72.71 (0.65) | 81.63 (0.55) | 71.55 (0.43) | 94.05 (0.49) | 91.77 (0.32) |
| KAA-SuperGAT   | 84.45 (0.23) | 73.08 (0.28) | 81.72 (0.43) | 74.01 (0.67) | 94.65 (0.78) | 92.20 (0.22) |
| HAT            | 83.67 (0.39) | 72.31 (0.23) | 79.67 (0.45) | 70.87 (0.65) | 93.77 (0.46) | 90.55 (0.34) |
| KAA-HAT        | 84.01 (0.40) | 72.55 (0.64) | 79.90 (0.42) | 73.55 (0.76) | 93.80 (0.37) | 91.65 (0.40) |
| NAGphormer     | 74.22 (0.89) | 63.68 (0.85) | 76.34 (0.76) | \            | 91.23 (0.78) | 90.04 (0.22) |
| KAA-NAGphormer | 77.01 (0.91) | 64.04 (0.67) | 78.30 (0.55) | \            | 91.42 (0.81) | 91.00 (0.45) |
| SGFormer       | 75.46 (0.78) | 67.79 (0.26) | 79.89 (0.64) | \            | 92.97 (0.44) | 91.06 (0.54) |
| KAA-SGFormer   | 77.45 (0.80) | 68.08 (0.77) | 80.03 (0.97) | \            | 93.08 (0.79) | 91.33 (0.59) |

Table. Accuracy (%) on node classification.



From the experimental table, we can observe that even for various high-performance attentive GNN models, KAA still achieves significant performance improvements. KAA consistently delivers performance gains across all scenarios and achieves remarkable results on certain datasets. For instance, KAA enhances HAT's performance on the ogbn-arxiv dataset by approximately 3%, which is quite impressive. In many cases, the performance gains brought by KAA surpass the performance differences between different models, highlighting the value of incorporating KAA. Furthermore, KAA is not at odds with advanced attentive GNN techniques and retains its effectiveness even when these techniques already exhibit strong performance. This is because existing methods primarily focus on identifying what constitutes important attention information, with advanced methods excelling at capturing such crucial information through the attention mechanism. In contrast, KAA enhances the actual construction process of the attention mechanism. As a result, regardless of the type of attention distribution being learned, KAA improves the success rate of accurately modeling that distribution.



**Reference**

[1] How to Find Your Friendly Neighborhood: Graph Attention Design with Self-Supervision.

[2] Hyperbolic Graph Attention Network.

[3] NAGphormer: A Tokenized Graph Transformer for Node Classification in Large Graphs.

[4] SGFormer: Simplifying and Empowering Transformers for Large-Graph Representations.

---

### Author Response · Authors · 2024-11-24
**Gentle Reminder: Discussion Phase Closing Soon**

Dear reviewers,
As the discussion phase is coming to a close, we kindly ask if our responses have fully addressed your questions and concerns. We truly appreciate your time and effort in reviewing our work and look forward to receiving your valuable feedback.

---

### Meta-Review · Area_Chair_Mpth · 2024-12-19

**Metareview:**

- Scientific Claims and Findings:
     -  This paper proposes using Kolmogorov-Arnold attention (KAA) to enhance attention-based GNN architectures, demonstrating improved performance on various graph prediction tasks and datasets.
     - It proposes a unified framework for scoring functions and introduces Maximum Ranking Distance (MRD) to theoretically analyse the expressive power of scoring functions in various graph attention mechanism.

- Strengths:
   - The integration of KAA into GNN architectures is new.
   - The theoretical analysis could be valuable to the research community.
   - The writing and presentation are clear and well-structured.

- Weaknesses:
    - The experimental results are somewhat weak and do not sufficiently demonstrate the effectiveness of KAA, potentially misaligning with the theoretical analysis.

- Most Important Reasons for Decision:
    - Considering the identified strengths, the work may be above the acceptance threshold for this conference. However, based on the AC's personal experience, the proposed KAA method might have limited practical value, as previous baseline models like GAT/GATv2 can achieve similar performance with careful hyperparameter tuning. I am recommending an acceptance, but I wouldn't mind if the paper gets rejected.

**Additional Comments On Reviewer Discussion:**

This paper initially received mixed reviews with scores of 5, 6, 6, and 3. After the rebuttal, Reviewer w7oB increased their score from 5 to 6, Reviewer kkV1 maintained their score of 6, Reviewer nWMV raised their score from 6 to 8, and Reviewer nAy3 increased their score from 3 to 5, and then to 6.

Overall, from the rebuttal discussion, the AC feels that while the contribution of this work may be technically solid, it is somewhat incremental and may not have significant practical value.

---

### Decision · Program_Chairs · 2025-01-22

Accept (Poster)